# Answer-Consistent Chain-of-Thought Reinforcement Learning for Multi-modal Large Language Models

## Abstract

Recent advances in large language models (LLMs) have demonstrated that reinforcement learning with verifiable rewards (RLVR) can significantly enhance reasoning abilities by directly optimizing correctness, rather than relying solely on supervised imitation. This paradigm has been extended to multimodal LLMs for complex video and image understanding tasks. However, while outcome-driven RL improves answer accuracy, it can inadvertently decouple the reasoning chain from the final answer, leading to situations where models produce inconsistency between the reasoning trace and final answer. In our experiments on multiple-choice visual question-answering tasks, the standard GRPO method yields only 79.7% consistency on MMVU between the reasoning steps and the chosen answers, indicating frequent mismatches between answers and reasoning. To this end, we propose Answer-Consistent REinforcement Learning (ACRE) that modifies the GRPO algorithm with an auxiliary consistency check. After the model generates a chain of thought and an initial answer for a given question, we shuffle the answer options and prompt the model again with the same reasoning trace to predict a second answer. We design a consistency-verification reward that grants a high reward only if both the original and the post-shuffle answers agree and are correct; otherwise, a lower reward is assigned accordingly. This mechanism penalizes reasoning-answer misalignment and discourages the model from relying on spurious patterns, such as option ordering biases. We evaluate ACRE on challenging Video Reasoning benchmarks and multimodal math reasoning benchmarks, achieving an average 2.2% and 1.5% improvement for Video Reasoning and Math Reasoning tasks over the GRPO baseline.

## 1 Introduction

The remarkable advancements in Large Language Models (LLMs) have been largely attributed to their emergent reasoning capabilities, often elicited through techniques like Chain-of-Thought (CoT) prompting. A pivotal step in refining these abilities has been the application of Reinforcement Learning (RL) with verifiable rewards, a paradigm that has demonstrated significant success in models like DeepSeek-R1 (Guo et al., 2025). This approach, which rewards the model for generating correct outcomes, has proven to be highly effective in enhancing the multi-step reasoning abilities of LLMs, particularly in domains where the final answer can be easily verified, such as mathematics and coding.

The success of RL in the text domain has naturally inspired researchers to explore its application in the realm of Multi-modal Large Language Models (MLLMs). The goal is to imbue these models, which can process and understand information from various modalities like images and videos, with sophisticated reasoning skills. Recent works such as Visual-RFT (Liu et al., 2025), Video-R1 (Feng et al., 2025), and Vision-R1 (Huang et al., 2025) have made significant strides in this direction. Visual-RFT extends reinforcement fine-tuning to visual perception tasks, demonstrating its data efficiency. Video-R1 adapts the Group Relative Policy Optimization (GRPO) algorithm to the video domain by introducing a temporal-aware reward mechanism. Vision-R1 employs GRPO with the hard formatting result reward function to gradually refine the model's ability to learn correct and complex reasoning processes on a 10K multimodal math dataset.

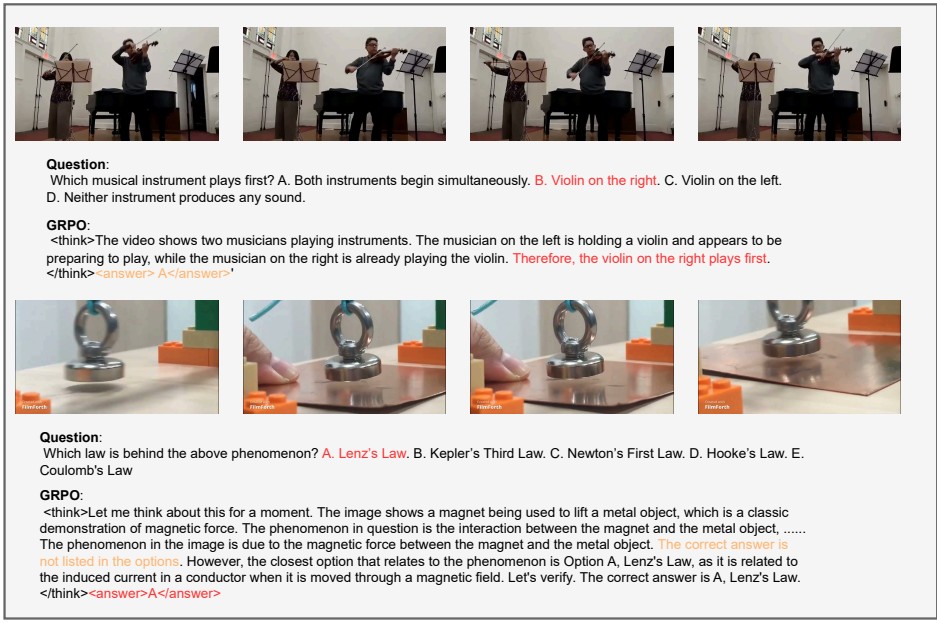

Figure 1: Reasoning-Answer inconsistency of GRPO models. Red denotes correct answer or reasoning trace and orange denotes flawed answer or reasoning trace. The top one is an example of Correct Reasoning but Wrong Answer, while the bottom one is an example of Wrong Reasoning but Correct Answer.

Despite these advances, our experiments reveal a subtle yet significant issue when applying RL to multi-choice image and video question-answering (QA) tasks. When we examine their reasoning trace and final answers, we observe an increasing trend of reasoning-answer mismatch. Specifically, of two undesirable states: either generating a correct and logical reasoning process but culminating in an incorrect final answer (denoted as CR–WA), or producing a flawed and inconsistent reasoning process that, by chance, leads to the correct answer (denoted as WR–CA). As illustrated in Fig.1, the top and bottom examples show CR–WA and WR–CA, respectively. Specifically, when analyzing Video-R1-7B's inference results on MMVU, there are 18.4% and 2.5% samples that belong to CR–WA and WR–CA, respectively. Specifically, in the CR–WA case, a negative advantage suppresses otherwise sound reasoning tokens and implicitly favors shorter, hedged traces, drifting the policy away from faithful step-by-step deduction. Conversely, in the WR–CA case, a positive advantage reinforces spurious shortcuts—such as option-index priors, positional heuristics, or visual/textual artifacts—that happened to yield the right letter, thereby increasing order sensitivity and brittle generalization. Together, these two modes amplify rationale–answer decoupling under outcome-only rewards. This "reasoning-answer inconsistency" suggests that the reward signal, which is based solely on the correctness of the final answer, may be inadvertently encouraging the model to find shortcuts or "guess" the right answer, rather than fostering a robust and reliable reasoning process. These behaviors undermine the trustworthiness and interpretability of the model, which are crucial for real-world applications.

To address this challenge, we propose ACRE: Answer-Consistent Chain-of-Thought Reinforcement Learning. Our method introduces a novel reward mechanism that explicitly promotes consistency between the reasoning process and the final answer. We modify the GRPO algorithm by introducing an auxiliary consistency check. During training, for a given multi-modal input and question, we first generate a response that includes a CoT reasoning process and a final answer. Then, to test the robustness of the generated reasoning, we shuffle the multiple-choice options and, using the original reasoning process, ask the model to generate a new answer. A maximal reward $r_{max}$ is given only if the answers from both the original and the shuffled-option settings are consistent with each other and match the ground truth. Otherwise, a lower reward is assigned. This second completion is solely for generating a more reliable reward signal for the first, complete generation (reasoning and answer), ensuring that the model is rewarded for producing a reasoning process that is not only correct but

also robust to variations in the answer space. By doing so, ACRE encourages the MLLM to develop a more grounded and reliable reasoning ability, resulting in more trustworthy and accurate responses in multimodal QA tasks.

To summarize, our contributions are listed as follows: **1)** We provide a comprehensive evaluation of the reasoning-answer inconsistency phenomenon in Multi-modal Reasoning Large Language Models. **2)** We curate a video-reasoning evaluation set on which GRPO-trained models are particularly prone to inconsistency errors, to help the community investigate implicit biases. **3)** We propose ACRE, a reinforcement learning algorithm based on GRPO that encourages more trustworthy reasoning traces. Comprehensive experiments in video reasoning benchmarks and multi-modal math reasoning benchmarks demonstrate our advantages over the GRPO baseline, surpassing 2.2% and 1.5% on average for video and math tasks, respectively, even out-performing the models post-trained on $28 \times$ samples.

## 2 RELATED WORK

**Reinforcement Learning for (Multi-modal) Large Language Models.** Reinforcement Learning for (Multi-modal) Large Language Models has evolved from preference-based alignment (e.g., DPO (Rafailov et al., 2023), ORPO (Hong et al., 2024)) to verifiable or outcome-based rewards that score answers with programmatic checks, unit tests, or reference solutions. DPO replaces Reinforcement Learning with Human Feedback (Bai et al., 2022; Ouyang et al., 2022) (RLHF)'s reward model and PPO loop with a closed-form objective, while ORPO further simplifies preference optimization without a reference model; both improve stability but still optimize preferences rather than correctness. Recent work formalizes and scales RLVR, often implemented with Group Relative Policy Optimization (GRPO) (Shao et al., 2024), and analyzes its effective loss and training dynamics for reasoning gains. Pure outcome-based methods may bring unexpected behavior. Existing literature has preliminarily demonstrated the mismatches between the reasoning traces and final answers (Lanham et al., 2023; Turpin et al., 2023). A complementary line of research focuses on enhancing reasoning via *process supervision*, which enforces the correctness of intermediate reasoning steps rather than solely the final outcome. Originating from text-based reasoning where step-level verification was shown to reduce logical errors (Lightman et al., 2024), this paradigm has been extended to the multimodal domain. For instance, VisualPRM (Wang & Others, 2025) utilizes fine-grained rewards to detect hallucinations, while recent work like Video-SALMONN-o1 (Sun & Others, 2025) introduces methods such as pDPO to integrate step-level supervision directly into the preference optimization framework, aligning the chain-of-thought with logical validity.

**Multi-modal Reasoning Large Language Models.** Inspired by advances in LLM reasoning, many studies have sought to enhance the reasoning capabilities of MLLMs. A primary strategy involves leveraging Chain-of-Thought (CoT) prompting to elicit step-by-step reasoning from the model (Wei et al., 2022). To further instill this capability, researchers have constructed specialized Supervised Fine-Tuning (SFT) datasets that contain detailed, step-level reasoning annotations. A prominent example is the ScienceQA dataset, which provides rich, explanatory rationales for multi-modal scientific questions (Lu et al., 2022). However, the CoT generated by these methods often follows a rigid, unidirectional inference path. This process frequently lacks the natural cognitive mechanisms inherent to human problem-solving, such as questioning, reflection, and inspection, which limits its effectiveness in complex, multi-step reasoning tasks. For instance, when faced with an ambiguous visual cue, the model cannot pause to ask a clarifying question or re-evaluate its initial interpretation. To address this gap, recent work has focused on developing more dynamic and iterative reasoning frameworks. These advanced models aim to emulate human-like cognition by incorporating self-correction and active exploration. For example, some frameworks enable MLLMs to critique and refine their own outputs in a feedback loop, thereby improving the accuracy and logical coherence of their reasoning paths (Madaan et al., 2023). Other approaches have endowed MLLMs with tool-use capabilities, allowing them to proactively seek external information or employ specialized models to verify intermediate steps, which is crucial for tasks requiring factual grounding and inspection (Lu et al., 2023b). By moving beyond static CoT, these methods aim to foster a more robust and flexible reasoning process, better equipping MLLMs to tackle the nuances of complex, real-world problems. Our work sits at the intersection of these lines: rather than relying solely on SFT-style CoT supervision or outcome-only RL, we shape the RL objective with an explicit consistency signal.

# 3 REASONING-ANSWER INCONSISTENCY IN MULTI-MODAL LARGE LANGUAGE MODELS

In this section, we dive into the Reasoning-Answer Inconsistency in the post-training of Multi-modal Large Language Models. Specifically, we are interested in Multi-modal Reasoning Large Language Models. That is, the reasoning MLLM, termed as $M_r$. When given a multi-modal input $x$ and a text query $q$, the model first generates a reasoning trace $o_{\text{think}}$ enclosed within `<think>` and `</think>` tags and then a final answer $o_{\text{ans}}$ enclosed within `<answer>` and `</answer>` tags.

A large amount of multimodal data is organized in the form of multiple-choice QA, not only because of its evaluation reliability and low labeling cost, but also because of its training convenience. It naturally supports binary rewards without external graders, making it ideal for GRPO-style post-training. However, models may unexpectedly learn option priors or lexical cues during reinforcement learning with pure outcome-based reward, leading to the reasoning-answer inconsistency. To systematically evaluate the phenomenon, we designed two tests, namely **CoT Answer Consistency Test** and **Option Shuffle Consistency Test**.

## 3.1 COT ANSWER CONSISTENCY TEST

Ideally, a reasoning MLLM should produce a final answer that is logically consistent with its chain-of-thought (CoT). Otherwise, the reported reasoning cannot be deemed reliable. We empirically observe that, after GRPO training, MLLMs more frequently generate final answers that contradict their own CoT. To quantify this effect, we adopt the LLM-as-Judge protocol. Let $f_{\text{judge}}$ denote the judge model and $P_{\text{judge}}$ its evaluation prompt. For each example $d \in D_{\text{test}}$, the model outputs a reasoning trace $o_{\text{think}}$ and a final answer $o_{\text{ans}}$. The CoT and Answer Consistency Rate (CACR) is defined as

$$\text{CACR} = \frac{1}{|D_{\text{test}}|} \sum_{d \in D_{\text{test}}} \mathbf{1}\Big[ f_{\text{judge}}\big( P_{\text{judge}}, o_{\text{think}}^{(d)}, o_{\text{ans}}^{(d)} \big) = \text{"consistent"} \Big], \tag{1}$$

where $\mathbf{1}[\cdot]$ is the indicator function that returns 1 if the judge deems the final answer consistent with the provided reasoning trace, and 0 otherwise. Please refer to the Appendix for the exact specification of $P_{\text{judge}}$.

## 3.2 OPTION SHUFFLING CONSISTENCY TEST

A strong reasoning MLLM should yield the same final answer when the user query is rephrased. Specifically, it should stay the same when the options are shuffled, provided that the multimodal evidence and the model's generated reasoning trace are held fixed. Let $\mathcal{S}(\cdot)$ be the option shuffling function applied to the query. The Option Shuffling Consistency Rate (OSCR) over a test set $D_{\text{test}}$ is defined as

$$\text{OSCR} = \frac{1}{|D_{\text{test}}|} \sum_{d \in D_{\text{test}}} \mathbf{1}\Big[ M_r\big( x^{(d)}, q^{(d)}, o_{\text{think}}^{(d)} \big) = M_r\big( x^{(d)}, \mathcal{S}(q^{(d)}), o_{\text{think}}^{(d)} \big) \Big], \tag{2}$$

where $\mathbf{1}[\cdot]$ is the indicator function that returns 1 if the two answers are identical and 0 otherwise. Higher values indicate better option shuffling consistency.

## 3.3 RESULTS ANALYSIS

We follow the same inference configuration as in Qwen2.5-VL, which is detailed in Sec.5. Table 1 illustrates the CACR on math reasoning benchmark, i.e., MathVista (Lu et al., 2023a), and the video reasoning benchmark, i.e., MMVU (Zhao et al., 2025). Table 2 demonstrates the OSCR on three video reasoning benchmarks. Since we use LLM-as-judge in computing CACR, to make the results more reliable, we additionally include human expert evaluations on a 50-sample subset of MathVista and MMVU, respectively. The results are denoted by MathVista-Human and MMVU-human, respectively.

**CACR patterns across training regimes.** We observe that CoT-SFT strengthens CACR while GRPO erodes CACR. Moving from instruction-only to CoT-SFT yields large CACR gains (e.g.,

Table 1: Results of CoT and Answer Consistency Rate (CACR) (%)

| Model | MathVista | MathVista-Human | MMVU | MMVU-Human |
|---|---|---|---|---|
| Qwen2.5-VL-7B-Instruct | 77.2 | 78.0 | 78.5 | 74.0 |
| Qwen2.5-VL-7B-CoT-SFT | 85.2 | 88.0 | 82.3 | 86.0 |
| Qwen2.5-VL-7B-CoT-SFT-GRPO | 81.3 | 82.0 | 79.7 | 82.0 |
| ACRE | 87.5 | 90.0 | 82.8 | 86.0 |

Table 2: Results of Option Shuffling Consistency Rate (OSCR) (%)

| Model | TempCompass | MMVU | VideoMME |
|---|---|---|---|
| ~~Qwen2.5-VL-7B-Instruct~~ | ~~88.8~~ | ~~97.0~~ | ~~86.4~~ |
| Gemini-2.5-pro* | 56.0 | 89.0 | 77.5 |
| Qwen2.5-VL-7B-CoT-SFT | 5.9 | 31.4 | 12.5 |
| Qwen2.5-VL-7B-CoT-SFT-GRPO | 10.4 | 49.8 | 25.6 |
| ACRE | 17.3 | 74.6 | 29.6 |

MathVista: $77.2 \rightarrow 85.2$, MMVU: $78.5 \rightarrow 82.3$), suggesting that supervised exposure to step-by-step traces teaches models to use their CoT as faithful rationales. While directly applying the COT prompts on Instruct models may not generate a reliable reasoning trace. GRPO lowers CACR relative to CoT-SFT across all columns (e.g., MathVista: $85.2 \rightarrow 81.3$, MMVU: $82.3 \rightarrow 79.7$), confirming that optimizing for correctness without explicitly coupling the rationale can decouple the decision head from the produced trace.

**OSCR patterns across training regimes.** Table 2 shows that the Qwen2.5-VL-7B-Instruct baseline attains the highest option–order robustness, achieving 88.8, 97.0, and 86.4 on TempCompass, MMVU, and VideoMME, respectively. This indicates that its final choices are tied to option content rather than index tokens. However, CoT-SFT collapses OSCR dramatically. For example, $88.81 \rightarrow 5.89$ on TempCompass. This reveals a strong index-binding effect: supervised traces often conclude with patterns like "thus the answer is (C)", which become brittle under permutation. Vanilla GRPO partially recovers robustness. For example, $12.5 \rightarrow 25.6$ on VideoMME, suggesting some reduction of index reliance but leaving substantial order sensitivity. Both results indicate that models after post-training somehow lose the power to retrieve the correct answer, given the correct reasoning trace.

## 4 ANSWER-CONSISTENT REINFORCEMENT LEARNING (ACRE)

While GRPO (Guo et al., 2025; Shao et al., 2024) effectively improves outcome accuracy, we observe that it can decouple the reasoning trace from the final answer in multimodal multiple-choice settings, leading to inconsistent behavior, as detailed in Sec.3. Granting a positive reward to WR-CA case and a negative reward to CR-WA is not desired. To address this, we propose **Answer-Consistent REinforcement Learning** (ACRE), a GRPO-compatible reward shaping scheme that explicitly aligns the generated chain-of-thought with the final answer (Fig. 2).

**Consistency check via Query Option Shuffling.** Given multimodal input $x$ (e.g., videos or images), a multiple-choice question $q$ with option set $\mathcal{O}$, and ground-truth answer $y \in \mathcal{O}$, the policy first produces a reasoning trace $t$ and an initial answer $a$:

$$t, a \sim \pi_\theta(\cdot \mid x, q).$$

We then apply a query option shuffling function $\mathcal{R}(\cdot)$ to obtain a rephrased question $S(q)$ while holding both $x$ and $t$ fixed, and re-prompt the model to produce a second answer $\tilde{a}$:

$$\tilde{a} \sim \pi_\theta(\cdot \mid x, \mathcal{S}(q), t).$$

Define the agreement indicator agree $= \mathbf{1}[a = \tilde{a}]$ and correctness indicators corr $= \mathbf{1}[a = y]$, $\widetilde{\text{corr}} = \mathbf{1}[\tilde{a} = y]$.

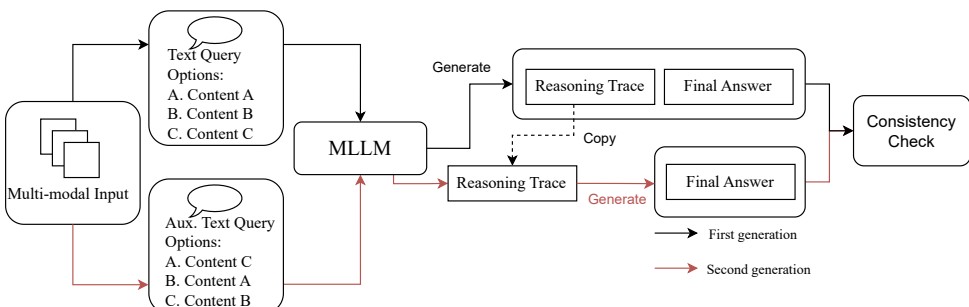

Figure 2: Overview of our proposed ACRE. Given a multi-modal input, the MLLM first generates a reasoning trace and a final answer (top path). We then feed the same reasoning trace back to the MLLM along with an auxiliary query where the answer options are shuffled (bottom path). The consistency between the final answers from both paths serves as a reward signal for reinforcement learning, encouraging the model to generate reasoning that is logically sound and independent of option positioning.

**Consistency-verification reward.** Let $r_i$ denote the base reward for trajectory $i$ that includes outcome correctness and format constraints following DeepSeek-R1 (Guo et al., 2025). ACRE adds a consistency term $r_c$ defined as

$$r_c = \begin{cases} \alpha_1, & \text{if agree} = 1 \text{ and corr} = \widetilde{\text{corr}} = 1, \\ \alpha_2, & \text{if agree} = 0 \text{ and } (\text{corr} + \widetilde{\text{corr}} = 1), \\ \alpha_3, & \text{if agree} = 1 \text{ and corr} = \widetilde{\text{corr}} = 0, \\ 0, & \text{otherwise.} \end{cases} \tag{3}$$

where $\alpha_1, \alpha_2, \alpha_3$ control the strength of positive reinforcement for answer-consistent reasoning and the penalty for misalignment, respectively. $\alpha_1$ is the standard reward for a perfect trajectory. $\alpha_2$ serves as partial credit: even if the consistency check fails, the model solved the original problem correctly. $\alpha_3$ encourages internal consistency. We refer the reader to the ablation study in Section.5, which empirically validates these choices. The final per-trajectory reward is

$$R_i = r_i + r_c. \tag{4}$$

This shaping grants a high bonus only when the reasoning trace yields stable and correct answers under query rephrasing, while discouraging reliance on spurious patterns such as option-order biases or reasoning–answer drift.

**Group-normalized advantages and policy update.** As in GRPO, we draw $G$ trajectories $\{o_i\}_{i=1}^{G}$ per prompt, compute rewards $\{R_i\}$ using equation 4, and normalize within the group:

$$A_i = \frac{R_i - \text{mean}(\{R_j\}_{j=1}^{G})}{\text{std}(\{R_j\}_{j=1}^{G}) + \varepsilon}. \tag{5}$$

The training objective augments GRPO with the consistency-shaped advantages:

$$\mathcal{J}_{ACRE}(\theta) = \mathbb{E}_{x,q,\{o_i\}} \left[ \frac{1}{G} \sum_{i=1}^{G} \left( \min\left( \frac{\pi_\theta(o_i \mid x,q)}{\pi_{\theta_{\text{old}}}(o_i \mid x,q)} A_i, \ \text{clip}\left( \frac{\pi_\theta(o_i \mid x,q)}{\pi_{\theta_{\text{old}}}(o_i \mid x,q)}, 1-\epsilon, 1+\epsilon \right) A_i \right) \right. \right.$$
$$\left. \left. - \beta \, \mathbb{D}_{\text{KL}}(\pi_\theta \| \pi_{\text{ref}}) \right) \right], \tag{6}$$

where $\epsilon$ is the PPO clipping parameter, $\beta$ scales the KL regularization to the reference model $\pi_{\text{ref}}$, and $\varepsilon$ is a small constant for numerical stability.

Table 3: Detailed Results Comparison. The train-data column indicates the amount of data used in the RL phase. We conduct experiments on video reasoning and multi-modal math task.

| Model | Train-data | Video Reasoning | | | | Math | | |
|---|---|---|---|---|---|---|---|---|
| | | VideoMME | MMVU | TempCompass | Avg. | MathVerse | MathVista | Avg. |
| Qwen2.5-VL-7B-Instruct | - | 0.503 | 0.598 | 0.708 | 0.603 | 0.421 | 0.685 | 0.553 |
| Qwen2.5-VL-7B-CoT-SFT | ACRE-9.2k | 0.528 | 0.613 | 0.692 | 0.611 | 0.456 | 0.680 | 0.568 |
| Qwen2.5-VL-7B-CoT-SFT-GRPO | ACRE-9.2k | 0.542 | 0.608 | 0.695 | 0.615 | 0.461 | 0.704 | 0.583 |
| Video-R1-7B | Video-R1-260k | **0.558** | 0.629 | **0.713** | 0.633 | 0.477 | 0.687 | 0.582 |
| **ACRE** | ACRE-9.2k | 0.545 | **0.656** | 0.710 | **0.637** | **0.481** | **0.715** | **0.598** |

## 5 EXPERIMENTS

### 5.1 SETUP

**Dataset Construction.** For the RL training dataset, we mix the Video-QA and Image-QA datasets. Specifically, we use Open-R1-Video-4.6k (Wang & Peng, 2025) for Video-QA. For Image-QA, we sample a 4.6k subset from the multiple-choice image QA data, including Math, Chart, OCR, Knowledge, and Spatial from Video-R1-260k (Feng et al., 2025). Together, we form a dataset of size 9.2k, which is named ACRE-9.2k.

**Implementation details**.We adopt Qwen2.5-VL-7B (Bai et al., 2025) as the base MLLMs. Similar to DeepSeekR1, the training process is conducted in two stages: SFT cold start followed by RL training. We directly use the SFT model provided by Video-R1(Feng et al., 2025) due to computational resource constraints. It is trained on the Video-R1-CoT-165k dataset, which contains chain-of-thought annotated samples derived from both image and video inputs. We denote this model as Qwen2.5-VL-7B-CoT-SFT. In the second stage, we further train the Qwen2.5-VL-7B-CoT-SFT model on ACRE-9.2k using GRPO and ACRE, the resulting model is named as Qwen2.5-VL-7B-CoT-SFT-GRPO and **ACRE** respectively. We also adopt the length-based reward to regulate the length of the model's output, introduced in Video-R1. Specifically, this mechanism aims to strike a balance between encouraging deeper reasoning and preventing overthinking. For each reasoning path $o_i$, if the predicted answer is correct and the response length falls within a predefined interval $[l_{\min}, l_{\max}]$, the model receives an additional reward $r_l = \omega$. Formally:

$$R_i = \begin{cases} R_i + \omega, & \text{if } o_i \text{ is correct and } l_{\min} \leq \text{len}(o_i) \leq l_{\max} \\ R_i, & \text{otherwise} \end{cases} \quad (7)$$

The hyper-parameter is set to be $\omega = 0.2$, $l_{min} = 320$ and $l_{max} = 512$.

For the CoT Answer Consistency Test, we adopt GPT-4o-mini as $f_{\text{judge}}$. For the Option Shuffling Consistency Test, we implement the query rephrase function as option shuffling function $S$. That is, for a multiple-choice query $q$, let $q = q_t + q_o$, where $q_t$ is the text query and $q_o$ is the option set. Then $S(q) = q_t + \text{derangement}(q_o)$, where derangement is a permutation of a set of elements where no element appears in its original position. This query rephrase function can be easily replaced with other functions if necessary.

**Evaluation Datasets** We evaluate our model on three video benchmarks and two multi-modal math benchmarks: VideoMME (Fu et al., 2025), MMVU (Zhao et al., 2025), TempCompass (Liu et al., 2024), MathVerse (Zhang et al., 2024), and MathVista (Lu et al., 2023a). For MathVerse and Math-Vista, evaluations are performed on their corresponding multiple-choice QA subset. For all evaluations, we follow the decoding configuration used in the official Qwen2.5-VL demo, with top_p = 0.001 and temperature = 0.01.

### 5.2 MAIN RESULTS

**Overall Performance.** As shown in Table 3, our experimental results across five benchmarks validate the effectiveness and data efficiency of ACRE for both video reasoning and multimodal math reasoning. Starting from the same CoT-SFT initialization, replacing vanilla GRPO with ACRE yields a +2.2 point improvement on the Video Reasoning Avg. (0.615 → 0.637) and a +1.5 point improvement on the Math Avg. (0.583 → 0.598). These averaged gains indicate that enforcing reasoning-answer agreement during training improves not only outcome accuracy but also the robustness of the decision stage after a chain-of-thought is produced. We further compare ACRE with

the GRPO baseline to assess generalization ability. Concretely, both methods are RL-finetuned only on video QA data, i.e., OpenR1-Video-4.6k, with **no** math QA exposure during the RL stage. We then evaluate the performance on two math reasoning benchmarks. Results in Table 4 show that ACRE surpasses GRPO on both MathVista (68.8 vs. 67.3) and MathVerse (45.7 vs. 44.5), yielding absolute gains of +1.5 and +1.2 points, respectively. Compared with Video-R1-7B, which is trained on a much larger dataset, the performance is still competitive. ( +0.4 on Video Reasoning and +1.6 on Math Reasoning)

**ACRE outperform GRPO in terms of CACR and OSCR.** As shown in Table.1 and Table.2, ACRE surpasses CoT-SFT baseline, achieving a much higher consistency between CoT and final answer. MathVista: **87.5** ( +2.3 vs. CoT-SFT, +6.2 vs. GRPO, +10.3 vs. Instruct); MMVU: **82.8** ( +0.5 vs. CoT-SFT / +3.1 vs. GRPO / +4.3 vs. Instruct). indicating little headroom under this cleaner split. Overall, ACRE attains the strongest or tied-strongest SCRs while retaining RL's accuracy benefits (Sec. 3).

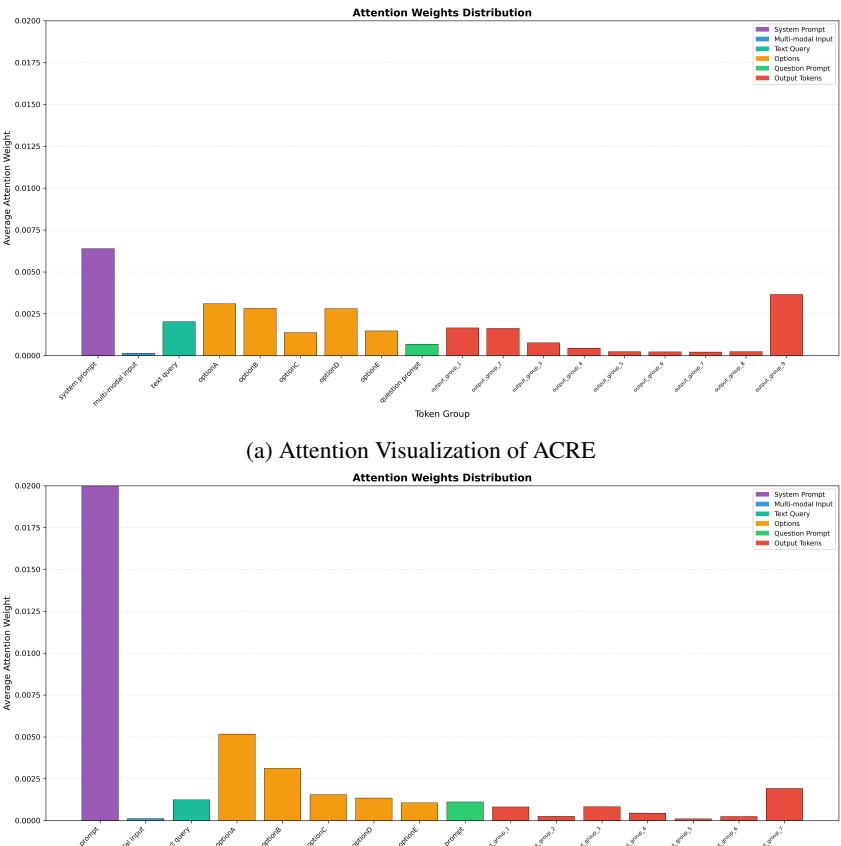

(a) Attention Visualization of ACRE

(b) Attention Visualization of GRPO

Figure 3: Attention Visualization Comparison between GRPO and ACRE

**Visualizations** Fig. 3 contrasts how attention is allocated across token groups for ACRE (top) and GRPO (bottom) when generating the final answer token, for a given Video QA whose answer is **A**. For the output reasoning tokens, we treat $n = 50$ tokens as a group and compute the average attention. Under GRPO, a dominant share of attention collapses onto the system prompt tokens (leftmost purple bar, $\approx 0.02$), while content-bearing regions—the question prompt, text query, and especially the options—receive comparatively weak weights. Unfortunately, although option A receives the most average attention across option sets, the GRPO model predicts it wrongly as **C**. This pattern is symptomatic of index/format shortcuts: the model keys on instruction or positional priors and then commits early to an option index, which explains its low robustness to option shuffling. In contrast,

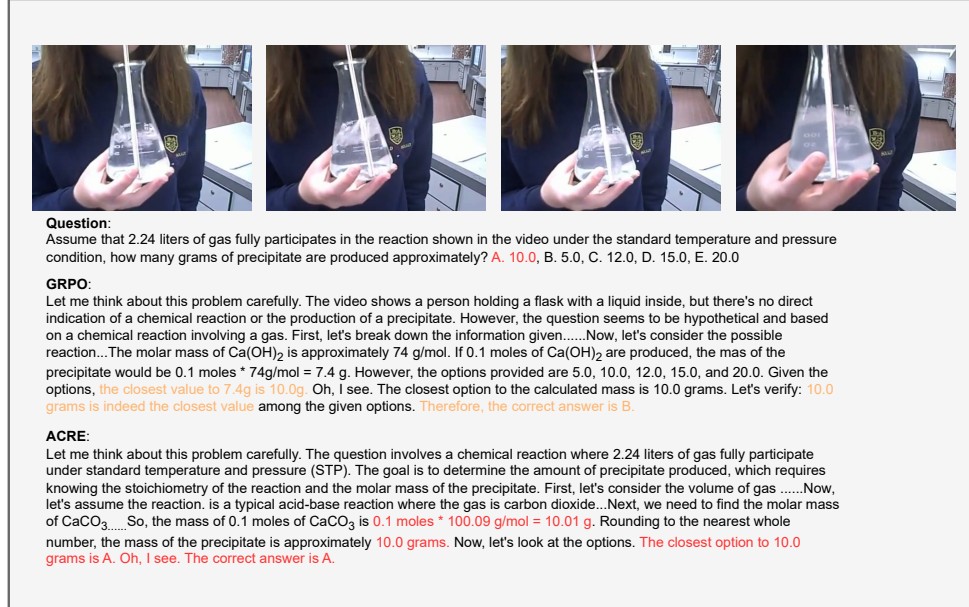

**Question**:
Assume that 2.24 liters of gas fully participates in the reaction shown in the video under the standard temperature and pressure condition, how many grams of precipitate are produced approximately? A. 10.0, B. 5.0, C. 12.0, D. 15.0, E. 20.0

**GRPO**:
Let me think about this problem carefully. The video shows a person holding a flask with a liquid inside, but there's no direct indication of a chemical reaction or the production of a precipitate. However, the question seems to be hypothetical and based on a chemical reaction involving a gas. First, let's break down the information given......Now, let's consider the possible reaction...The molar mass of Ca(OH)$_2$ is approximately 74 g/mol. If 0.1 moles of Ca(OH)$_2$ are produced, the mas of the precipitate would be 0.1 moles * 74g/mol = 7.4 g. However, the options provided are 5.0, 10.0, 12.0, 15.0, and 20.0. Given the options, the closest value to 7.4g is 10.0g. Oh, I see. The closest option to the calculated mass is 10.0 grams. Let's verify: 10.0 grams is indeed the closest value among the given options. Therefore, the correct answer is B.

**ACRE**:
Let me think about this problem carefully. The question involves a chemical reaction where 2.24 liters of gas fully participate under standard temperature and pressure (STP). The goal is to determine the amount of precipitate produced, which requires knowing the stoichiometry of the reaction and the molar mass of the precipitate. First, let's consider the volume of gas ......Now, let's assume the reaction. is a typical acid-base reaction where the gas is carbon dioxide...Next, we need to find the molar mass of CaCO$_3$......So, the mass of 0.1 moles of CaCO$_3$ is 0.1 moles * 100.09 g/mol = 10.01 g. Rounding to the nearest whole number, the mass of the precipitate is approximately 10.0 grams. Now, let's look at the options. The closest option to 10.0 grams is A. Oh, I see. The correct answer is A.

Figure 4: Visualizations of GRPO and ACRE. Red denotes correct answer or reasoning trace and orange denotes flawed answer or reasoning trace.

Table 4: Generalization Performance (%) and Training GPU hours

| Model | Train data | MathVista | MathVerse | Training GPU hours |
|---|---|---|---|---|
| Qwen2.5-VL-CoT-SFT-GRPO | OpenR1-Video-4.6k | 0.673 | 0.445 | 4.5 |
| ACRE | OpenR1-Video-4.6k | 0.688 | 0.457 | 5.6 |

ACRE redistributes attention away from the system prompt and toward the options (multiple orange bars), and the output tokens (red), with a smoother spread across option spans and a pronounced rise near the final output groups. The resulting profile indicates that the model is attending to the content of alternatives while forming the final choice, rather than relying on spurious index cues—a behavior consistent with our higher CACR and improved OSCR. Fig.4 shows model performance comparison between GRPO and ACRE.

## 5.3 ABLATIONS

**Training Time**   Since we need to forward twice to compute the consistency reward, this inevitably brings extra computational overhead. In Table 4, we compare the total training GPU hours of GRPO and ACRE on OpenR1-Video-4.6k training data. The training GPU hours show a slight increase from 4.5 to 5.6 (+24%), which is acceptable because in the second forward pass we only generate the final answer token rather than re-decoding the full CoT. Moreover, this overhead can be further reduced: in our current implementation, we do not reuse the KV cache from the first pass.

**Hyperparameters**   We ablate the consistency-shaping coefficients $\alpha_1, \alpha_2, \alpha_3$ (Sec. 4). Fixing $\alpha_1 = 1$, we first vary $\alpha_2 \in \{1.0, 0.9, 0.8, 0.7\}$ with $\alpha_3 = 0.3$. The corresponding performance on MMVU are 0.645, **0.656**, 0.637, 0.638. This suggests that cases where the model's two-pass answers disagree but exactly one is correct should receive a reward close to $\alpha_1$ yet strictly smaller: $\alpha_2 = 0.9$ strikes the best balance, whereas over-rewarding ($\alpha_2 = 1.0$) or under-rewarding ($\alpha_2 \leq 0.8$) both degrade performance. Next, fixing $\alpha_2 = 0.9$, we vary $\alpha_3 \in \{0.0, 0.3, 0.5\}$, obtaining 0.643, **0.656**, 0.629. A moderate positive $\alpha_3 = 0.3$ (reward for agreement when both answers are incorrect) helps stabilize learning—likely by encouraging internally consistent traces while other signals steer correctness—whereas either no shaping ($\alpha_3 = 0.0$) or too much shaping ($\alpha_3 = 0.5$) harms results. Overall, the best setting is $(\alpha_1, \alpha_2, \alpha_3) = (1, 0.9, 0.3)$, indicating that

mild encouragement of agreement and near-top reward for "one-correct" disagreement yield the strongest trade-off between robustness and accuracy.

## 5.4 DISCUSSION

While ACRE is designed for multiple-choice questions (MCQ), it can be easily adapted for open-ended data during RL training. Here we propose two practical strategies.

**Mixed-reward strategy** In practice, ACRE does not need to be the only reward mechanism: For MCQs, we can directly apply the ACRE consistency reward to strictly enforce reasoning alignment. For open-ended QA, math, and code tasks, we can use outcome-based rewards (e.g., correct answer extraction, unit tests, code execution) or reference-based metrics (ROUGE/BLEU). This allows the model to benefit from ACRE's consistency enforcement without requiring every sample to be in MCQ format.

**Automated conversion via distractor generation** To directly apply ACRE's consistency check to open-ended data, we can use an automated distractor generation approach. A strong MLLM generates plausible but incorrect options (distractors) from the ground-truth answer, temporarily converting an open-ended question into an MCQ for training. Specifically, for an instruction tuning sample $d = (x, q, r)$ (where $x$ is the visual input, $q$ is the query, and $r$ is the ground-truth response), we feed the original query $q$ and the ground-truth answer $r$ to GPT-4o-mini. We instruct it to generate 3 *hard negative* candidates $(r_1, r_2, r_3)$. These candidates must be topically relevant to the query but contain substantial factual flaws relative to the ground truth (e.g., changing the action, object, or attribute described). Then, we conduct query reformatting by rewriting the original query $q$ into a selection task $q'$ (e.g., "Choose the most appropriate description..."). And therefore, the new data point becomes an MCQ tuple $d' = (x, q', \{r, r_1, r_2, r_3\})$, which is then used for ACRE training. We conduct a preliminary experiment, detailed in the Appendix. We leave ACRE's broader experiments with a larger, more diverse dataset for future work.

## 6 CONCLUSION AND FUTURE WORK

**Conclusion** This paper studied the reasoning–answer inconsistency that emerges when outcome-only reinforcement learning is applied to multimodal, multiple-choice reasoning. We first diagnosed the problem using two complementary tests—the *CoT and Answer Consistency Rate* (CACR) and the *Option Shuffling Consistency Rate* (OSCR)—and showed that vanilla GRPO improves answer accuracy yet erodes consistency between the generated chain-of-thought (CoT) and the final answer. To address this, we introduced Answer-Consistent REinforcement Learning (ACRE), a GRPO-compatible reward shaping scheme that enforces shuffle-invariant agreement conditioned on correctness. Concretely, ACRE reuses the model's own reasoning trace while perturbing option order, and it allocates reward according to a four-way consistency signal. Across five benchmarks spanning video and multimodal math reasoning, ACRE yields consistent gains over GRPO (e.g., +2.2 points on the Video Reasoning Avg. and +1.5 points on the Math Reasoning Avg.) while restoring or surpassing CoT alignment as reflected by CACR and improving robustness as reflected by OSCR. These results indicate that coupling outcome optimization with explicit consistency verification produces models that both reason more faithfully and decide more robustly.

**Future Work** We presently implement query rephrasing via option shuffling, which explicitly promotes robustness and consistency in the model's reasoning. The framework is agnostic to the specific perturbation and can readily incorporate alternative rephrasing strategies—for example, prompting an LLM to produce semantically equivalent paraphrases in varied forms.

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

# A APPENDIX

## A.1 LLM USAGE STATEMENT

We clarify that the use of LLMs in this study is restricted to writing assistance, specifically for grammar correction and enhancing readability. No LLM was involved in the research design, experimental execution, or data analysis. The authors take full responsibility for the content of the manuscript, including any text generated or polished by the LLM. We have ensured that the LLM-generated text adheres to ethical guidelines and does not contribute to plagiarism or scientific misconduct.

## A.2 PROMPTS USED IN SEC.3

The prompts are included in the supplementary.zip file as the judge_prompt.md

## A.3 CODES

Codes are provided in the supplementary.zip file as ACRE_code.zip

