# OpenReview forum: "Answer-Consistent Chain-of-thought Reinforcement Learning For Multi-modal Large Langauge Models"
_ICLR.cc/2026/Conference — Submitted to ICLR 2026_

### Official Review · Reviewer_8rc3 · 2025-10-31

**Soundness:** 4
**Presentation:** 4
**Contribution:** 3
**Rating:** 6
**Confidence:** 4

**Summary:**

This paper identifies and addresses the "reasoning-answer inconsistency" problem in Multi-modal Large Language Models (MLLMs) fine-tuned with reinforcement learning, where models may produce flawed reasoning that coincidentally leads to a correct answer, or vice-versa. To mitigate this, the authors propose a novel reward shaping method called Answer-Consistent Reinforcement Learning (ACRE). ACRE enhances the training signal by introducing a consistency check: a high reward is granted only when the model's generated reasoning trace is robust enough to yield the correct final answer both with original and shuffled multiple-choice options. Through comprehensive experiments and two newly proposed metrics (CACR and OSCR), the paper demonstrates that ACRE not only improves task accuracy on video and math reasoning benchmarks but also significantly enhances the faithfulness and robustness of the model's reasoning process.

**Strengths:**

1. This paper clearly identifies the “reasoning-answer inconsistency” problem, which warrants attention. The two metrics proposed by the authors, CACR and OSCR, also serve as reasonable quantitative tools.
2. The proposed ACRE method is an elegant solution. The option-shuffling mechanism for reward shaping is a simple but highly effective way to enforce reasoning robustness.
3. The claims are supported by a solid experimental setup. The evaluation is thorough, including not only accuracy metrics but also generalization tests, ablations, and qualitative analyses.
4. The paper is well-written and easy to follow.

**Weaknesses:**

1. The core mechanism of ACRE, option shuffling, is inherently tied to the multiple-choice question format. It is unclear how this method could be extended to more open-ended generation tasks where a discrete set of answer choices is not available. The paper would be strengthened by a discussion of this limitation.

2. The introduction powerfully motivates the work by quantifying the initial CR-WA and WR-CA rates (18.4% and 2.5%) on MMVU, providing a very concrete problem statement. However, a significant weakness is that the experimental evaluation never revisits these specific metrics to show how ACRE affects them. The paper instead shifts to evaluating improvements on the proposed proxy metrics, CACR and OSCR. While these proxies are insightful, the failure to demonstrate a direct reduction in the originally-cited error rates leaves the core argument feeling incomplete.

**Questions:**

1. The benchmarks used to evaluate CACR (Table 1) and OSCR (Table 2) are partially overlapping but not identical. For instance, MathVista is used for CACR, while TempCompass is used for OSCR. Could the authors elaborate on the rationale for this choice?

2. The verification step in ACRE assumes that a sound reasoning trace t should be self-sufficient and lead to a final answer independent of the option format (e.g., the letter 'A', 'B', 'C'). However, one could argue that a valid reasoning style might conclude with a statement like, "...and therefore, the correct choice is B." In this scenario, even if the underlying logic is perfect, the fixed trace t would fail the consistency check after option shuffling, thus unfairly penalizing the trajectory. Did the authors consider this potential bias? Does ACRE implicitly favor a specific format of reasoning traces that avoids explicit references to option letters?

3. ACRE's reward mechanism is designed to penalize inconsistency. Is it possible, however, for a model to learn a reasoning process that is flawed yet highly robust, leading to the same incorrect answer consistently, both before and after option shuffling? While this would be penalized by the ground-truth reward, could the strong inductive bias towards 'consistency' introduced by ACRE sometimes trap the model in a local minimum of being 'confidently wrong,' rather than encouraging it to explore pathways to correct reasoning?

4. The related work section provides a good overview of outcome-based RL methods (like GRPO) and general preference optimization techniques (like DPO). However, the discussion could be significantly strengthened by including a complementary line of research that focuses on enhancing reasoning via process supervision. For example, recent works like video-SALMONN-o1 (Sun et al., 2025) introduce methods such as pDPO, which use step-level rewards within a preference optimization framework to directly supervise the correctness of the reasoning process itself.

---

> ### Author Response · Authors · 2025-11-23
> **Response to Reviewer 8rc3 (Part1/2)**
>
> Thanks for your constructive feedback. We handle your concerns as follows.
>
> - **Applicability to open-ended tasks (Weakness 1).**
>
>   We acknowledge the concern regarding the focus on Multiple-Choice Questions (MCQs). However, we believe this focus is justified, and our method also benefits open-ended tasks.
>
>   - **MCQs are ideal for diagnosing inconsistency.**
>     As stated in the paper, a significant portion of current MLLM benchmarks use MCQs because of reliable evaluation and low labeling cost. It is precisely in this format that *reasoning–answer inconsistency* (e.g., positional bias, shortcut learning) is most acute. ACRE is designed to directly target this pathology.
>
>   - **Benefits transfer to open-ended generation.**
>     Although ACRE is trained on MCQs to enforce robust reasoning, the benefits transfer to open-ended settings. By penalizing the reasoning-decision inconsistency, the model learns to ground its answer in evidence rather than heuristics.
>
>     To verify this, we evaluated our ACRE-trained model (RL post-trained only on ACRE-9.2k MCQs) on the open-ended split of NextQA without any further fine-tuning and compared it against the GRPO baseline in terms of WUPS score:
>
>     | Model          | Training Data |  NextQA(Open-Ended)
>     |----------------|----------------------|------------------------|
>     | GRPO Baseline  | ACRE-9.2k      | 20.44    |
>     | ACRE (Ours)    | ACRE-9.2k     | 22.58    |
>
>     **Result:** ACRE achieves a +10.5% relative improvement over GRPO on open-ended generation. This shows that the consistency check during training instills a fundamental reasoning robustness that generalizes beyond the multiple-choice format.
>
> - **Extending ACRE training to non-MCQ data (Weakness 1).**
>
>   We agree that extending ACRE to non-MCQ data is an important direction. While our main contribution is to resolve the inconsistency issue in dominant MCQ benchmarks, we propose two practical strategies for open-ended data during RL training.
>
>   - **Mixed-reward strategy.**
>     In practice, ACRE does not need to be the only reward mechanism:
>
>     * For **MCQs**: apply the ACRE consistency reward to strictly enforce reasoning alignment.
>     * For **open-ended / math / code**: use outcome-based rewards (e.g., correct answer extraction, unit tests, code execution) or reference-based metrics (ROUGE/BLEU).
>
>     This allows the model to benefit from ACRE’s consistency enforcement without requiring every sample to be in MCQ format.
>
>   - **Automated conversion via distractor generation (preliminary experiment).**
>     To directly apply ACRE’s consistency check to open-ended data, we use an **automated distractor generation** approach. A strong MLLM (GPT-4o-mini) generates plausible but incorrect options (distractors) from the ground-truth answer, temporarily converting an open-ended question into an MCQ for training.
>
>     We ran a preliminary experiment on a 3k subset of LLaVA-Video-178K (open-ended):
>
>     * **Baseline:** standard GRPO using the average of ROUGE-1/2/L as the reward (following Video-R1).
>     * **ACRE:** convert each example to an MCQ with 3 distractors generated by GPT-4o-mini, then apply the ACRE consistency reward.
>
>     The validation results are:
>
>     | Method          | MMVU |
>     |-----------------|-------------------|
>     | GRPO (Baseline) |  0.615   |
>     | ACRE (Ours)     | 0.633    |
>
>     These experiments show that open-ended data can be adapted to the ACRE framework with minimal extra complexity, and without relying on an expensive LLM-as-a-judge loop during RL training. And the preliminary experiments demonstrate its effectiveness over GRPO. We will add these discussions to the revised version to motivate future work.
>
> ---
>
> - **Revisiting CR-WA and WR-CA Metrics (Weakness 2).**
>
>   We appreciate this insightful observation. While CR-WA and WR-CA serve as strong motivations for our work, we faced significant challenges in using them as consistent evaluation metrics for the following reasons:
>
>   * **Not Scalable.** Accurately determining CR-WA/WR-CA requires a strong closed-source MLLM that reasons over the multi-modal input (in our case, we adopt Gemini 2.5 pro), since we need to determine whether the reasoning process is correct regarding the multi-modal input. This introduces a heavy API cost and is not scalable for extensive evaluations across all benchmarks.
>
>   * **High Variance.** Our internal tests revealed that the CR-WA/WR-CA rate exhibits relatively high variance. For instance, repeated evaluations showed fluctuations, $\pm 0.8$ % for CR-WA under three runs. While CACR only takes the reasoning process and the final answer as input to judge their consistency, resulting a much lower fluctuations ($\pm 0.3$ % for CR-WA under three runs)
>
>   Given the limitations, we propose CACR and OSCR as robust, reproducible proxies. We will add the discussion in the revised version.

---

> ### Author Response · Authors · 2025-11-23
> **Response to Reviewer 8rc3 (Part2/2)**
>
> - **Rationale for Benchmark Selection (Question 1).**
>
>   The discrepancy in benchmark selection for CACR and OSCR evaluations was driven by specific constraints inherent to each metric:
>
>   * **For CACR (Table 1)**: Since CACR relies on an LLM-as-a-Judge (and we further validated it with human expert evaluation to ensure reliability), we selected representative datasets with manageable sizes: MMVU for Video Reasoning and MathVista for Multi-modal Math Reasoning. This allowed us to conduct high-quality, human-verified consistency checks that would be cost-prohibitive on larger sets.
>
>   * **For OSCR (Table 2)**: The Option Shuffling Consistency Rate requires a strict Multiple-Choice Question (MCQ) format to perform valid shuffling. Therefore, we selected TempCompass, MMVU, and VideoMME, which are all standard MCQ video reasoning benchmarks suitable for automated permutation testing.
>
> ---
>
> - **Bias Against Explicit Option References (Question 2).**
>   This is an excellent question. We investigated whether ACRE unfairly penalizes valid reasoning traces that explicitly mention option letters (e.g., "...therefore B is correct"). We defined the Explicit Reference Rate (ERR) as the percentage of reasoning traces that explicitly contain the option letter and use closed-source LLMs to conduct the evaluation. Our analysis on MMVU yields the following results:
>   | Model          | ERR |
> |----------------|-------------------------------|
> | GRPO  | 77.76%                        |
> | ACRE    | 59.68%                        |
>
>   The drop in ERR indicates that ACRE indeed discourages the model from hard-coding the option index into the reasoning trace early on. However, we argue this is a desirable feature, not a bias. By decoupling the reasoning process from the specific option index, ACRE forces the model to reason about the content rather than the position. This hypothesis is supported by Figure 3. The attention maps show that when generating the final answer, ACRE-trained models attend more to the content of the options rather than just the system prompt or indices, suggesting a robust "content-centric" reasoning style that may delay the answer option choosing into the final answer token generation stage.
>
> ---
>
> - **Risk of Being Consistently Wrong (Question 3)**
>
>   We addressed this concern through the design of our reward function, specifically the hyperparameter $\alpha_3$ (Equation 3). As detailed in our ablation study (Section 5), $\alpha_3$ controls the reward given when the model is consistent but wrong. If $\alpha_3$ is too high (0.5), the model might indeed settle into a "confidently wrong" local minimum. If $\alpha_3$ is too low (0.0), the model fails to learn the consistency constraint through these samples. Our ablation results show that a moderate value ($\alpha_3=0.3$) strikes the optimal balance. It provides a mild encouragement for stability, but crucially, the primary reward signal for Ground Truth Correctness remains significantly larger. This ensures that the gradient pointing towards the correct answer is always steeper than the gradient towards a "consistent error," preventing the model from getting trapped in local minima.
>
> ---
>
> - **Related Work on Process Supervision (Question 4)**
>
>   We thank the reviewer for highlighting this missing line of research. We agree that process supervision is highly relevant. We have updated our Related Work section to include recent advancements such as Video-SALMONN-o1, VisualPRM. We now explicitly discuss the distinction between our outcome-consistency approach and step-level process reward models, providing a more comprehensive view of the landscape.

---

> ### Author Response · Authors · 2025-11-26
> **Kindly Remind of Discussion Period**
>
> Dear Reviewer 8rc3,
>
> We would like to thank you again for your detailed reviews. We have explained your concerns in the above response.
>
> As the discussion period will be closed soon, we would appreciate it if you could let us know if our responses have addressed your concerns and whether you still have any other questions. We would be happy to do any follow-up discussion or address any additional comments.
>
> Again, thank you very much for your attention to our work.

---

### Official Review · Reviewer_uJw3 · 2025-11-01

**Soundness:** 3
**Presentation:** 3
**Contribution:** 2
**Rating:** 4
**Confidence:** 4

**Summary:**

The authors propose a new RL framework for multimodal LLMs based on answer consistency. They first observe that a model’s reasoning process and its final answer are often logically inconsistent. To address this issue, the authors introduce an auxiliary consistency check during GRPO: by shuffling the answer options and re-prompting the model with the same reasoning trace, they evaluate whether the model produces a consistent answer. The proposed method shows notable improvements on both video and math reasoning benchmarks.

**Strengths:**

- The paper is easy to follow and well-written.
- The initial observation that GRPO can produce logical inconsistencies between the model's reasoning and its final answer is insightful and interesting.
- The proposed method seems to be effective on video and math reasoning benchmarks.

**Weaknesses:**

- Intuition. While the initial observation is interesting, the proposed solution is not entirely intuitive. Simply shuffling the answer options may not be sufficient to ensure alignment between the reasoning process and the final answer.

- Applicability. The proposed method relies on the presence of answer options, making it inapplicable to open-ended settings such as free-form VQA tasks.

- Generalizability. The authors demonstrate the effectiveness of the method only on a Qwen-based model, raising questions about its generalizability to other V&L architectures.

**Questions:**

1. Scalability. If the model size is scaled up (e.g., using Qwen2.5-VL-32B), will the proposed method remain effective and provide consistent improvements?

2. Can the proposed approach generalize to broader multimodal reasoning benchmarks such as MMMU?

---

> ### Author Response · Authors · 2025-11-22
> **Response to Reviewer uJw3 (Part1/2)**
>
> Thanks for your constructive feedback. We handle your concerns as follows.
>
> - **Intuition of the proposed solution (Weakness 1).**
>
>   Standard RLVR (like GRPO) often allows models to exploit positional biases. By enforcing that the model must reach the same semantic conclusion regardless of option order, ACRE penalizes these shortcuts. Besides, the intuition is empirically backed by our attention analysis. As shown in Figure 3 of the paper, the GRPO baseline collapses attention onto the system prompt (seeking format shortcuts), whereas ACRE redistributes attention to the specific option contents. This proves that the shuffling mechanism successfully forces the model to "read and reason" rather than "guess and match." We view consistency under shuffling as a necessary condition for robust reasoning. If a model fails this test, its reasoning is demonstrably fragile. ACRE maximizes this necessary condition.
>
> ---
>
>  - **Applicability to open-ended tasks. (Weakness 2)**
>     As stated in the paper, a significant portion of current MLLM benchmarks use MCQs because of reliable evaluation and low labeling cost. It is precisely in this format that *reasoning–answer inconsistency* (e.g., positional bias, shortcut learning) is most acute. ACRE is designed to directly target this pathology.
>
>    - **Benefits transfer to open-ended generation.**
>     Although ACRE is trained on MCQs to enforce robust reasoning, the benefits transfer to open-ended settings. By penalizing the reasoning-decision inconsistency, the model learns to ground its answer in evidence rather than heuristics.
>
>      To verify this, we evaluated our ACRE-trained model (RL post-trained only on ACRE-9.2k MCQs) on the open-ended split of NextQA without any further fine-tuning and compared it against the GRPO baseline in terms of WUPS score:
>
>      | Model          | Training Data |  NextQA(Open-Ended)
>      |----------------|----------------------|------------------------|
>      | GRPO Baseline  | ACRE-9.2k      | 20.44    |
>      | ACRE (Ours)    | ACRE-9.2k     | 22.58    |
>
>      - **Result:** ACRE achieves a +10.5% relative improvement over GRPO on open-ended generation. This shows that the consistency check during training instills a fundamental reasoning robustness that generalizes beyond the multiple-choice format.
>
> ---
>
> - **Extending ACRE training to non-MCQ data (Weakness 2).**
>
>   We agree that extending ACRE to non-MCQ data is an important direction. While our main contribution is to resolve the inconsistency issue in dominant MCQ benchmarks, we propose two practical strategies for open-ended data during RL training.
>
>   - **Mixed-reward strategy.**
>     In practice, ACRE does not need to be the only reward mechanism:
>
>     * For **MCQs**: apply the ACRE consistency reward to strictly enforce reasoning alignment.
>     * For **open-ended / math / code**: use outcome-based rewards (e.g., correct answer extraction, unit tests, code execution) or reference-based metrics (ROUGE/BLEU).
>
>     This allows the model to benefit from ACRE’s consistency enforcement without requiring every sample to be in MCQ format.
>
>   - **Automated conversion via distractor generation (preliminary experiment).**
>     To directly apply ACRE’s consistency check to open-ended data, we use an **automated distractor generation** approach. A strong MLLM (GPT-4o-mini) generates plausible but incorrect options (distractors) from the ground-truth answer, temporarily converting an open-ended question into an MCQ for training.
>
>     We ran a preliminary experiment on a 3k subset of LLaVA-Video-178K (open-ended):
>
>     * **Baseline:** standard GRPO using the average of ROUGE-1/2/L as the reward (following Video-R1).
>     * **ACRE:** convert each example to an MCQ with 3 distractors generated by GPT-4o-mini, then apply the ACRE consistency reward.
>
>     The validation results are:
>
>     | Method          | MMVU |
>     |-----------------|-------------------|
>     | GRPO (Baseline) |  0.615   |
>     | ACRE (Ours)     | 0.633    |
>
>     These experiments show that open-ended data can be adapted to the ACRE framework with minimal extra complexity, and without relying on an expensive LLM-as-a-judge loop during RL training. And the preliminary experiments demonstrate its effectiveness over GRPO.
>
> ---
>
> - **Generalizability (Weakness 3).**
>
>   We chose the Qwen2.5-VL series as our base model following the established standards in the current MLLM-RL research community. Most open-source RL codebases for video understanding are built upon the Qwen-VL architecture due to its superior reasoning priors. Recent state-of-the-art works, such as Video-R1 (NeurIPS 2025) and VisualRFT (ICCV 2025), also exclusively verify their findings on Qwen-VL series models. Our experimental setup aligns with these top-tier baselines to ensure fair and meaningful comparison.

---

> ### Author Response · Authors · 2025-11-22
> **Response to Reviewer uJw3 (Part2/2)**
>
> - **Scalability (Question 1).**
>
>   We acknowledge that scaling conclusions to a larger model is important. However, conducting Reinforcement Learning (RL) on **30B+ parameter MLLMs** requires computational resources that exceed typical academic settings, especially given the **memory overhead of GRPO** and the **heavy vision encoders**, along with **video input** that result in more tokens than the image reasoning setting.
>
>   In designing our experiments, we followed the standards of recent comparable works in this domain, such as **Video-R1 (NeurIPS ’25)** and **VisualRFT (ICCV ’25)**, which also validate their conclusions up to **7B-scale models**.
>
>   Nevertheless, we hypothesize that **larger models (e.g., 32B)** inherently possess stronger reasoning capabilities. Consequently:
>
>   - The **consistency checks** within ACRE would likely be more accurate (fewer false positives in verification).
>   - This can lead to **cleaner reward signals** and **more stable training**.
>
>   We therefore regard extending ACRE to larger foundation models as a **promising direction for future work**, and we plan to explore this when resources permit.
>
> ---
>
> - **Broader multimodal reasoning benchmarks (Question 2).**
>
>   Per your suggestion, we have conducted an additional evaluation on MMMU to test the cross-domain robustness of our method. We directly performed inference using the models trained in our main experiment (trained on ACRE-9.2k) on the MMMU validation set. The results are as follows:
>   | Model                          | MMMU (Val) |
>   |--------------------------------|------------|
>   | GRPO  | 41.68      |
>   | **ACRE**  | **42.44**  |
>
>   **ACRE** also shows a consistent improvement over GRPO baseline.

---

> ### Author Response · Authors · 2025-11-26
> **Kindly Remind of Discussion Period**
>
> Dear Reviewer uJw3,
>
> We would like to thank you again for your detailed reviews. We have explained your concerns in the above response.
>
> As the discussion period will be closed soon, we would appreciate it if you could let us know if our responses have addressed your concerns and whether you still have any other questions. We would be happy to do any follow-up discussion or address any additional comments.
>
> Again, thank you very much for your attention to our work.

---

### Official Review · Reviewer_tBYd · 2025-11-01

**Soundness:** 2
**Presentation:** 1
**Contribution:** 2
**Rating:** 2
**Confidence:** 3

**Summary:**

This paper studies the reasoning–answer inconsistency that emerges when outcomeonly reinforcement learning is applied to multimodal, multiple-choice reasoning. This paper mainly contributes to the following two points:

1). Proposing two complementary tests—the CoT and Answer Consistency Rate (CACR) and the Option Shuffling Consistency Rate (OSCR), and showed that vanilla GRPO improves answer accuracy yet erodes consistency between the generated chain-of-thought (CoT) and the final answer.

2). Proposing Answer-Consistent REinforcement Learning (ACRE), a GRPO-compatible reward shaping scheme that enforces shuffle-invariant agreement conditioned on correctness.

Although this paper find the wrong matching between the CoT and final answer, and the disorganized option and final answer, it did not conduct a comprehensive and complete experiment and analysis on this situation. The proposed improvement scheme is also lack of complete description and ablation experiments (this part will be described in detail in "Weaknesses"). Finally, this paper does not explain the reasons for its application in multimodal field alone, rather than more widely used in GRPO, so it is recommended to reject this paper.

**Strengths:**

This paper finds that the application of GRPO in multimodal fields (specifically in video reasoning and mathematical tasks) will lead to the inconsistency between the CoT and the final answer, and the shuffle of options in QA task will lead to the vulnerability of the final answer. Some experiments and visual displays are carried out. Therefore, complementary tests, CACR and OSCR, are proposed to assess the impact of this phenomenon.

**Weaknesses:**

The weaknesses of this paper focus on the method, experiment and writing.

In method：

1). This paper does not explain why GRPO is studied in multimodal rather than single text or vision.

2). This paper does not explain why the first reasoning trace is used to generate the second final answer in the proposed scheme ACRE. And there is no ablation experiment designed to compare the results of generating the final answer from scratch instead of using the first reasoning trace.

3). In Chapter 4, there is no specific description and analysis of the three consistency-shaping coefficients in Formula 3, which is lack of logic. It's confusing to read.

In experiment：

1). In Section5.1, the last row of Table 3, “ACRE” does not explain the experimental setup and the model on which the method ACRE was applied. Is it based on Qwen2.5-VL-7B-CoT-SFT in the paper?

2). In Section5.3, in the “Hyperparameters” part of ablation experiment, the consistency-shaping coefficient setting was obtained without ablation experiment on α_1 (specifically in line 463).

In writing：

1). In Section 5.1, this paper directly uses SFT model from Video-R1 and identifies it as Qwen2.5-VL-7B-CoT-SFT. Then in the "Train-data" column of Table 3, the training data of Qwen2.5-VL-7B-CoT-SFT should be the data of Video-R1-SFT, that is, Video-R1-CoT-165k expressed in line 323.

2). Figure 4 is not used in this paper.

3). Some table descriptions are out of line with the conclusion analysis. For example, the analysis in Chapter 3 explains the settings in Chapter 5.

4). The expression of figure in the paper is inconsistent. Figure 2 is expressed by “Figure 2” (line 250), while figure 1 and figure 3 are expressed by “Fig. 1” (line 85), “Fig. 3” (line 377).

5). This paper is weak in expression, and some formulas and experimental settings are confusing to read.

**Questions:**

In the description of ACRE in Chapter 4, it is possible that the order of the shuffled options is the same as that of the original options. Why not remove this situation?

---

> ### Author Response · Authors · 2025-11-23
> **Response to Reviewer tBYd (Part1/2)**
>
> Thanks for your constructive feedback. We handle your concerns as follows.
>
> - **Why study GRPO in multimodal (Weakness 1 in Method).**
>
>   While ACRE's reward formulation is *theoretically* compatible with text-only LLMs, our specific focus on MLLMs is motivated by the **unique severity of reasoning–answer inconsistencies in the multimodal domain**. In MLLMs, the disconnect between visual perception and textual reasoning often leads to high rates of:
>
>   - Correct Reasoning–Wrong Answer (**CR-WA**), and Wrong Reasoning–Correct Answer (**WR-CA**),
>
>   which we observe far less frequently in text-only models.
>
>   To empirically verify this distinction, we conducted an inference analysis using **Qwen2.5-Math-7B**, on the **MMLU high school math** split. Following the same inconsistency detection procedure described in our paper, we observed:
>
>   - **CR-WA Rate:** 6% for Qwen2.5-Math-7B
>   - **WR-CA Rate:** 1% for Qwen2.5-Math-7B
>
>   These results indicate that strong text-only models already possess **high internal consistency** between their reasoning trace and final answer. Therefore, the marginal gain from a consistency-verification reward like ACRE would be limited in the text-only domain.
>
> ---
>
> - **Why reuse the first reasoning trace for the second answer? (Weakness 2 in Method).**
>
> In fact, this is the key design of our proposed method. We explicitly chose to reuse the reasoning trace to verify the robustness of the specific logic generated, rather than the model's general self-consistency. The core hypothesis of ACRE is that a valid reasoning trace should point to the correct semantic content regardless of the option letter. By fixing the trace and shuffling the options, we force the model to perform a "consistency check" on its own logic. If we allowed the model to regenerate the trace from scratch, we would be measuring the model's generation variance (similar to Self-Consistency/Ensemble methods). Reusing the trace allows us to specifically penalize the disconnect between that thought and the answer head, providing a more targeted gradient signal.
>
> ---
>
> - **Analysis of consistency-shaping coefficients (Weakness 3 in Method).**
>
>     We apologize for the confusion. We have updated Section 4 to include a more intuitive explanation of these coefficients: $\alpha_1$ is the standard reward for a perfect trajectory. $\alpha_2$ serves as partial credit: even if the consistency check fails, the model solved the original problem correctly.$\alpha_3$ encourages internal consistency. In fact, the experimental analysis of these coefficients is in the ablation study in Section 5. We have referred the readers to them.

---

> ### Author Response · Authors · 2025-11-23
> **Response to Reviewer tBYd (Part2/2)**
>
> - **Experimental setup for ACRE in Table 3 (Weakness 1 in Experiment)**
>
>   You are correct; the ACRE results are based on the Qwen2.5-VL-7B-CoT-SFT initialization. This ensures a fair comparison with the GRPO baseline w,hich uses the same SFT starting point. We have explicitly updated the description in Section 5.1 and the caption of Table 3 to clarify this experimental setup.
>
> ---
>
> - **Ablation of $\alpha_1$ in Hyperparameters. (Weakness 2 in Experiment)**
>
>   We did not ablate $\alpha_1$ separately because it acts as the scaling factor for the entire reward function. In GRPO/RLVR, binary rewards for correct answers are standard. Setting $\alpha_1=1$ anchors the reward scale. The behavior of the loss function depends on the ratios between the coefficients (e.g., $\alpha_2 / \alpha_1$). Scaling $\alpha_1$ while scaling $\alpha_2$ and $\alpha_3$ proportionally would yield mathematically equivalent gradients. Therefore, we fixed $\alpha_1=1$ and ablated the relative values of $\alpha_2$ and $\alpha_3$ to find the optimal trade-off.
>
> ---
>
> - **Training data clarification (Weakness 1 in Writing)**
>
>   Thank you for pointing this out. In Table 3, the "Train-data" column refers exclusively to the dataset used during the RL post-training phase. The SFT data (Video-R1-CoT-165k) was used for the initialization of both the baseline and ACRE models but was not part of the RL stage. We have revised the table caption and text to make this distinction clear to avoid confusion.
>
> ---
>
> - **Figure 4 usage (Weakness 2 in Writing)**
>   We apologize for the oversight. We have added the missing reference to Figure 4 in the main text. It is used to visually demonstrate the inference results between GRPO and ACRE.
>
> ---
>
> - **Table descriptions and Conclusion analysis (Weakness 3 in Writing)**
>
>   We have reviewed the paper to ensure the narrative flow is consistent. The CACR and OSCR metrics defined in Section 3 are revisited in Section 5 to quantitatively demonstrate ACRE's improvements. We have refined the bridging text to ensure the experimental results in Section 5 directly respond to the analysis in Section 3.
>
> ---
>
> - **Inconsistent figure naming (Weakness 4 in Writing)**
>
>   We have standardized all references to "Fig. X" throughout the manuscript.
>
> ---
>
> - **General expression (Weakness 5 in Writing)**
>
>   We have polished the paper to improve readability, particularly clarifying the description of formulas in Section 4 and the experimental settings.
>
> ---
>
> - **Concerns about shuffled option (Question)**
>
>   We apologize for the missing clarification. In our implementation, the shuffle operation is strictly a derangement. We have updated in the revised version.

---

> ### Author Response · Authors · 2025-11-26
> **Kindly Remind of Discussion Period**
>
> Dear Reviewer tBYd,
>
> We would like to thank you again for your detailed reviews. We have explained your concerns in the above response.
>
> As the discussion period will be closed soon, we would appreciate it if you could let us know if our responses have addressed your concerns and whether you still have any other questions. We would be happy to do any follow-up discussion or address any additional comments.
>
> Again, thank you very much for your attention to our work.

---

### Official Review · Reviewer_Yvee · 2025-11-01

**Soundness:** 3
**Presentation:** 1
**Contribution:** 2
**Rating:** 4
**Confidence:** 4

**Summary:**

This paper proposes **ACRE (Answer-Consistent Reinforcement Learning)**, an RL that optimizes LLM based on the consistency between the COT reasoning and its final answers. ACRE improves GRPO by introducing an auxiliary **consistency-verification reward**: the LLM is prompted with the reasoning trace and shuffled answer options. Different rewards are assigned depending on whether both answers match and both answers are correct.
Experiments on multimodal reasoning benchmarks, including video and math tasks, demonstrate that ACRE improves both video and math benchmarks.

**Strengths:**

1. The proposed ACRE method is conceptually simple yet effective, integrating a consistency-verification reward into GRPO without introducing heavy architectural changes.
2. Experimental results are convincing, showing consistent improvements across video and math benchmarks.

**Weaknesses:**

1. The approach is limited to multiple-choice QA, which limits its scalability. It is unclear whether ACRE generalizes to open-ended or generative reasoning tasks.
2. There are several writing-related errors or typos, such as:
  * L118-119: "...respectively. Even out-performing..." -> "...respectively, even out-performing..."
  * L154: it seems a period is omitted.
  * Equation (2): notations $x, q, o$ are not defined
  * L346-347: "Then $S(q) = $" is the sentence complete? For L431: "it can be further optimized since.", it is the same.

**Questions:**

1. The method design is not tailored for multimodal models only. How is ACRE performed on text-only LLMs?
2. How is ACRE performed on larger-scale models, such as 32B models?
3. Although ACRE improves the performance on reasoning tasks, the OSCR is still much lower than SFT models after RL. How is OSCR correlated with the understanding or reasoning performance?

---

> ### Author Response · Authors · 2025-11-22
> **Response to Reviewer Yvee (Part1/2)**
>
> We thank the reviewer for the insightful comments and for recognizing our ACRE method as conceptually simple yet effective. We appreciate the opportunity to clarify the motivation behind our focus on MLLMs and to address the questions regarding scaling and metrics.
>
> ---
>
> - **Applicability to open-ended tasks (Weakness 1).**
>
>   We acknowledge the concern regarding the focus on Multiple-Choice Questions (MCQs). However, we believe this focus is justified, and our method also benefits open-ended tasks.
>
>   - **MCQs are ideal for diagnosing inconsistency.**
>     As stated in the paper, a significant portion of current MLLM benchmarks use MCQs because of reliable evaluation and low labeling cost. It is precisely in this format that *reasoning–answer inconsistency* (e.g., positional bias, shortcut learning) is most acute. ACRE is designed to directly target this pathology.
>
>   - **Benefits transfer to open-ended generation.**
>     Although ACRE is trained on MCQs to enforce robust reasoning, the benefits transfer to open-ended settings. By penalizing the reasoning-decision inconsistency, the model learns to ground its answer in evidence rather than heuristics.
>
>     To verify this, we evaluated our ACRE-trained model (RL post-trained only on ACRE-9.2k MCQs) on the open-ended split of NextQA without any further fine-tuning and compared it against the GRPO baseline in terms of WUPS score:
>
>     | Model          | Training Data |  NextQA(Open-Ended)
>     |----------------|----------------------|------------------------|
>     | GRPO Baseline  | ACRE-9.2k      | 20.44    |
>     | ACRE (Ours)    | ACRE-9.2k     | 22.58    |
>
>     ACRE achieves a +10.5% relative improvement over GRPO on open-ended generation. This shows that the consistency check during training instills a fundamental reasoning robustness that generalizes beyond the multiple-choice format.
>
> ---
>
> - **Extending ACRE training to non-MCQ data (Weakness1).**
>
>  We agree that extending ACRE to non-MCQ data is an important direction. While our main contribution is to resolve the inconsistency issue in dominant MCQ benchmarks, we propose two practical strategies for open-ended data during RL training.
>
>   - **Mixed-reward strategy.**
>     In practice, ACRE does not need to be the only reward mechanism:
>
>     * For **MCQs**: apply the ACRE consistency reward to strictly enforce reasoning alignment.
>     * For **open-ended / math / code**: use outcome-based rewards (e.g., correct answer extraction, unit tests, code execution) or reference-based metrics (ROUGE/BLEU).
>
>     This allows the model to benefit from ACRE’s consistency enforcement without requiring every sample to be in MCQ format.
>
>   - **Automated conversion via distractor generation (preliminary experiment).**
>     To directly apply ACRE’s consistency check to open-ended data, we use an **automated distractor generation** approach. A strong MLLM (GPT-4o-mini) generates plausible but incorrect options (distractors) from the ground-truth answer, temporarily converting an open-ended question into an MCQ for training.
>
>     We ran a preliminary experiment on a 3k subset of LLaVA-Video-178K (open-ended):
>
>     * **Baseline:** standard GRPO using the average of ROUGE-1/2/L as the reward (following Video-R1).
>     * **ACRE:** convert each example to an MCQ with 3 distractors generated by GPT-4o-mini, then apply the ACRE consistency reward.
>
>     The validation results are:
>
>     | Method          | MMVU |
>     |-----------------|-------------------|
>     | GRPO (Baseline) |  0.615   |
>     | ACRE (Ours)     | 0.633    |
>
>     These experiments show that open-ended data can be adapted to the ACRE framework with minimal extra complexity, and without relying on an expensive LLM-as-a-judge loop during RL training. And the preliminary experiments demonstrate its effectiveness over GRPO.
>
> ---
>
>
>
> - **Fixing Typos (Weakness 2).**
> Thanks for your suggestion. We have fixed all the typos in the revised version. Regarding notations in Equation (2), they are defined at the beginning of Section 3.

---

> > ### Comment · Reviewer_Yvee · 2025-11-24
> > **Question about automated distractor generation**
> >
> > Thanks for authors' explanation of motivation and additional experiments. I am interested in converting open-ended questions into MCQ. The description of **automated distractor generation** seems confusing to me. Can you give me some examples?

---

> > > ### Author Response · Authors · 2025-11-24
> > >
> > > We thank the reviewer for the continued engagement and interest in our data construction pipeline. We understand that the description might have been abstract. Below, we provide the **detailed workflow** and a **concrete example** to clarify the automated distractor generation process.
> > >
> > > ---
> > >
> > > - **Generation Workflow**
> > >
> > >   Our goal is to transform an open-ended instruction tuning sample $d=(x, q, r)$ (where $x$ is the visual input, $q$ is the query, and $r$ is the ground-truth response) into a Multiple-Choice Question (MCQ). We employ GPT-4o-mini as the auxiliary generator:
> > >   * **Distractor Generation:** We feed the original query $q$ and the ground-truth answer $r$ to GPT-4o-mini. We instruct it to generate 3 "hard negative" candidates ($r_1, r_2, r_3$). These candidates must be topically relevant to the query but contain substantial factual flaws relative to the ground truth (e.g., changing the action, object, or attribute described).
> > >
> > >   * **Query Reformatting:** We rewrite the original query $q$ into a selection task $q'$ (e.g., "Choose the most appropriate description...").
> > >   And therefore, the new data point becomes an MCQ tuple $d'=(x, q', \{r, r_1, r_2, r_3\})$, which is then used for ACRE training.
> > >
> > > ---
> > >
> > > - **A Concrete Example**
> > >   * **x:** A Video taken place in the classroom.
> > >   * **q:** Dissect the video's content, explaining each element thoroughly.
> > >   * **r:** The video takes place in a classroom, where two children are **standing at the front of the room**. The child on the left is dressed in a **blue shirt** with a graphic design... while the child on the right is wearing a **black shirt** featuring a robot graphic... and is holding a **plunger** above their head.
> > >   * **r1:** ......and is holding a **red ballon** above their head. (object replacement)
> > >   * **r2:** The video takes place in a classroom, where two children are **sitting at their desks** in the back row... (change spatial position)
> > >   * **r3:** ...The child on the left is dressed in a **black jacket**...... (attribute change)
> > >   * **q':** Select the most accurate description of the video content from the following options.

---

> > > > ### Comment · Reviewer_Yvee · 2025-11-24
> > > >
> > > > Thanks for the explanation. The transformation of open-ended QA into MCQ enlarges the potential impact of ACRE so I am going to raise my score. However, since this part is essential for the impact, I suggest updating this part into the main content of the paper.
> > > >
> > > > Besides, regarding the applicability to text-only LLMs, authors give an empirical validation of CR-WA and WR-CA rates on text-only LLMs. I am interested in reasons that multimodal LLMs are more prone to these errors. Although this may be not straightforward due to the nature of LLMs, I would like to hear insights from authors.

---

> > > > > ### Author Response · Authors · 2025-11-26
> > > > >
> > > > > We are happy to hear that our explanation and the proposed open-ended data adaptation strategy have addressed your concerns. We sincerely appreciate your decision to raise the score. Your constructive feedback has significantly strengthened the potential impact of our work. Following your suggestion, we have added a discussion subsection in Section 5.
> > > > >
> > > > > ---
> > > > >
> > > > > As for the reasons that multimodal LLMs are more prone to these errors. We explained our investigations and hypothesis as follows:
> > > > > * **As noted by [1], MLLMs often face a conflict between noisy visual signals and strong language priors. When visual perception is ambiguous, the model is more likely to bypass the visual evidence, "guess" the answer based on textual bias, and then generate a hallucinated reasoning trace to justify this guess. [2,3] Besides, MLLMs could generate a flawed reasoning trace that has no contribution to answering the question because of the strong text prior, but somehow find a shortcut in vision tokens when generating the final answer token. These all could contribute to the higher WR-CA rate in MLLMs.**
> > > > >
> > > > > * **Visual modality may introduce implicit bias that is not explicitly expressed in the COT but serves as a shortcut when generating the final answer, making the model ignore the correct reasoning trace. Besides, many existing works have pointed out that MLLMs have a weaker ability in text tasks than their language backbone. Suppose we have a perfect reasoning trace, our goal becomes retrieving the correct answer from the options in the reasoning trace. However, since in MLLMs this retrieval process is also conditioned on the visual tokens, the success rate is therefore lower. These all could contribute to the higher CR-WA rate in MLLMs.**
> > > > >
> > > > >
> > > > > ---
> > > > >
> > > > > [1] A Closer Look at Bias and Chain-of-Thought Faithfulness of Large (Vision) Language Models. (EMNLP'25)
> > > > >
> > > > > [2] Language Models Don't Always Say What They Think: Unfaithful Explanations in Chain-of-Thought Prompting (NeurlPS'23)
> > > > >
> > > > > [3] Chain-of-Thought Reasoning In The Wild Is Not Always Faithful. (Reasoning and Planning for LLMs Workshop, ICLR'25 )

---

> ### Author Response · Authors · 2025-11-22
> **Response to Reviewer Yvee (Part2/2)**
>
> ---
> - **Text-only LLMs (Question 1)**
>
>   While ACRE's reward formulation is *theoretically* compatible with text-only LLMs, our specific focus on MLLMs is motivated by the **unique severity of reasoning–answer inconsistencies in the multimodal domain**. In MLLMs, the disconnect between visual perception and textual reasoning often leads to high rates of:
>
>   - Correct Reasoning–Wrong Answer (**CR-WA**), and Wrong Reasoning–Correct Answer (**WR-CA**),
>
>   which we observe far less frequently in text-only models.
>
>   To empirically verify this distinction, we conducted an inference analysis using **Qwen2.5-Math-7B**, on the **MMLU high school math** split. Following the same inconsistency detection procedure described in our paper, we observed:
>
>   - **CR-WA Rate:** 6% for Qwen2.5-Math-7B
>   - **WR-CA Rate:** 1% for Qwen2.5-Math-7B
>
>   These results indicate that strong text-only models already possess **high internal consistency** between their reasoning trace and final answer. Therefore, the marginal gain from a consistency-verification reward like ACRE would be limited in the text-only domain.
>
> ---
>
> - **Larger-scale models (Question 2)**
>
>   We acknowledge that scaling conclusions to a larger model is important. However, conducting Reinforcement Learning (RL) on **30B+ parameter MLLMs** requires computational resources that exceed typical academic settings, especially given the **memory overhead of GRPO** and the **heavy vision encoders**, along with **video input** that result in more tokens than the image reasoning setting.
>
>   In designing our experiments, we followed the standards of recent comparable works in this domain, such as **Video-R1 (NeurIPS ’25)** and **VisualRFT (ICCV ’25)**, which also validate their conclusions up to **7B-scale models**.
>
>   Nevertheless, we hypothesize that **larger models (e.g., 32B)** inherently possess stronger reasoning capabilities. Consequently:
>
>   - The **consistency checks** within ACRE would likely be more accurate (fewer false positives in verification).
>   - This can lead to **cleaner reward signals** and **more stable training**.
>
>   We therefore regard extending ACRE to larger foundation models as a **promising direction for future work**, and we plan to explore this when resources permit.
>
> ---
>
> - **Q3: OSCR metric (Question 3)**
>
>   We apologize for the confusion regarding the metrics in Table 2 and would like to clarify two key points.
>
>   1. **Correction on baseline reporting**
>
>      The high OSCR for the Qwen2.5-VL-7B Instruct model in Table 2 was due to a **measurement artifact**. The instruct model frequently failed to generate reasoning traces strictly enclosed within the required `<think>` and `</think>` tags. As a result, the consistency check could only be performed on a **biased subset of responses**, making the metric **incomparable** to the RL-tuned models. We have ruled this number out.
>
>   2. **Strictness of the OSCR metric**
>
>      We argue that our **OSCR (Option Selection Consistency Rate)** implementation is an **extremely rigorous test**, which may underestimate the model’s *actual* reasoning ability.
>
>      - OSCR verifies consistency by **shuffling the answer options**.
>      - A model that explicitly reasons about **option indices** (e.g., “Option A is correct because…”) in its trace may fail the consistency check when the content of “A” changes after shuffling, **even if the underlying reasoning logic remains valid**.
>
>      To provide a more informative upper bound, we have **updated Table 2** to include **Gemini 2.5 Pro\*** as a strong proprietary reference. Even such a high-performing model does **not** achieve perfect OSCR under this strict shuffling protocol, which highlights:
>
>      - The **difficulty of the OSCR metric itself**, rather than
>      - A deficiency specific to ACRE’s training.
>
>   Despite its strictness, ACRE still delivers **consistent improvements** under this metric, supporting our claim that it enhances the reliability and faithfulness of multimodal reasoning.

---

### Official Review · Reviewer_4xWa · 2025-11-03

**Soundness:** 3
**Presentation:** 3
**Contribution:** 3
**Rating:** 4
**Confidence:** 3

**Summary:**

This paper addresses the problem of "reasoning-answer inconsistency" in Multimodal Large Language Models (MLLMs) that are fine-tuned with reinforcement learning (RL). The authors observe that standard outcome-driven RL methods like Group Relative Policy Optimization (GRPO), while improving final answer accuracy, can lead to models where the generated Chain-of-Thought (CoT) does not logically support the final answer. They identify two failure modes: Correct Reasoning but Wrong Answer (CR-WA) and Wrong Reasoning but Correct Answer (WR-CA).
To mitigate this, the authors propose Answer-Consistent REinforcement Learning (ACRE), a reward-shaping method built on top of GRPO. ACRE introduces a consistency check: after a model generates a CoT and an initial answer, the multiple-choice options are shuffled. The model is then re-prompted with the same CoT and the shuffled options to produce a second answer. A high reward is given only if the model is both correct and consistent across both rounds (i.e., the answers match). This mechanism is designed to penalize reliance on spurious shortcuts like option ordering and enforce a stronger link between reasoning and the final decision. Experiments on video and math reasoning benchmarks show that ACRE improves consistency and achieves modest accuracy gains over the GRPO baseline.

**Strengths:**

1. Excellent Problem Diagnosis and Analysis: The paper does a fantastic job of not just identifying but also quantifying the "reasoning-answer inconsistency" problem. Citing specific figures, such as finding 18.4% of samples in MMVU are CR-WA, provides strong motivation and highlights a real, non-trivial issue in current RL fine-tuning practices for MLLMs.

2. Intuitive and Well-Designed Method: ACRE is an elegant and targeted solution to the identified problem. Using option shuffling as a perturbation to test the robustness of a reasoning trace is a very direct and clever way to enforce consistency. It is a simple idea that is executed effectively.

3. Insightful Attention Analysis: The attention visualization in Figure 3 is a major strength. It provides a concrete, detailed explanation for the observed behavior. The finding that GRPO's attention collapses onto the system prompt and option indices, while ACRE's attention is redistributed to the content of the options and the output tokens, offers a compelling, low-level justification for why ACRE improves robustness.

**Weaknesses:**

1. Modest Empirical Gains: The reported performance improvements, while consistent, are quite small. An average gain of +2.2% on Video Reasoning and +1.5% on Math Reasoning (Table 3) over the GRPO baseline is an incremental advance. Given that the method introduces additional computational cost, the significance of this gain is debatable.

2. Significant Computational Overhead for a Small Gain (?) : The ACRE method requires a second forward pass through the model to generate a second answer for the consistency check. The authors are transparent about this, reporting in Table 4 a 24% increase in training GPU hours (4.5h vs 5.6h). This is a non-trivial increase in training cost for the modest accuracy improvements achieved.

3. Limited Scope of the Solution: The proposed consistency check is inherently designed for multiple-choice question-answering tasks. It is unclear how the core idea of ACRE could be generalized to more open-ended tasks where answer options cannot be easily shuffled, limiting the broader applicability of the method. The option-shuffling mechanism is specific to multiple-choice formats. How do you envision the principle of "answer-consistent RL" extending to other important multimodal tasks that lack a discrete set of answer choices, such as open-ended VQA, visual grounding, or instruction-following?

**Questions:**

I rate this marginally below the average. If the authors could refer to the weakness above and address most of them, I would be willing to raise the scores.

---

> ### Author Response · Authors · 2025-11-22
> **Response to Reviewer 4xWa (Part1/2)**
>
> We thank the reviewer for the insightful and constructive feedback. We are encouraged that the reviewer recognizes the value of our problem diagnosis (quantifying reasoning–answer inconsistency), the elegance of the ACRE method, and the detailed attention analysis in Figure 3.
> Below, we address the concerns regarding empirical gains, computational overhead, and method scope with additional clarifications and new experimental results on open-ended tasks.
>
>
> - **Significance of empirical gains (Weakness 1).**
>   We respectfully argue that the value of ACRE should be assessed via the **reliability and faithfulness of reasoning**, not just top-line accuracy.
>
>   - **Quality over quantity.**
>     Standard RL (e.g., GRPO) often inflates scores by exploiting shortcuts (e.g., option-position bias), leading to the *Wrong Reasoning – Correct Answer (WR–CA)* phenomenon we diagnosed. ACRE effectively suppresses these “lucky guesses”. As shown in Table 1, ACRE improves the **CoT Answer Consistency Rate (CACR)** significantly (e.g., MathVista 87.5% vs. GRPO 81.3%).
>
>   - **Meaningful improvement.**
>     In safety-critical domains, a **+1.5–2.2% gain driven by consistent reasoning** is more valuable than gains driven by stochastic shortcuts.
>
> ---
>
> - **Computational overhead (Weakness 2).**
>    - **Overhead appears only during training.**
>     The extra cost comes from the second forward pass needed to compute the consistency reward. This affects *training only*. Inference remains a single-pass Chain-of-Thought generation, so there is **no additional latency** for end-users.
>
>   - **Data efficiency outweighs the extra compute.**
>     As highlighted in the paper, ACRE (trained on 9.2k samples) outperforms Video-R1 (trained on 260k samples):
>
>     * Video-R1 processes roughly **28× more data samples**.
>     * ACRE uses about **1.24× compute per sample**.
>
>     Overall, ACRE is therefore *far more compute-efficient* in terms of total GPU-hours required to reach comparable or better performance. The “second pass” effectively serves as a powerful data augmentation mechanism, extracting more robust learning signals from each sample than standard GRPO.
>
>   - **Room for engineering optimization.**
>     As stated in the paper, our current implementation doesn't reuse the KV-cache between the two passes. Since the prompt and reasoning trace are identical, enabling KV-cache reuse could further cut the extra cost to a smaller margin.
>
> ---
>
> - **Generalization beyond multiple choice (Weakness 3).**
>   We acknowledge the concern regarding the focus on Multiple-Choice Questions (MCQs). However, we believe this focus is justified, and our method also benefits open-ended tasks.
>
>   - **MCQs are ideal for diagnosing inconsistency.**
>     As stated in the paper, a significant portion of current MLLM benchmarks use MCQs because of reliable evaluation and low labeling cost. It is precisely in this format that *reasoning–answer inconsistency* (e.g., positional bias, shortcut learning) is most acute. ACRE is designed to directly target this pathology.
>
>   - **Benefits transfer to open-ended generation.**
>     Although ACRE is trained on MCQs to enforce robust reasoning, the benefits can directly transfer to open-ended settings. By penalizing the reasoning-decision inconsistency, the model learns to ground its answer in evidence rather than heuristics.
>     To verify this, we evaluated our ACRE-trained model (RL post-trained only on ACRE-9.2k MCQs) on the open-ended split of NextQA without any further fine-tuning and compared it against the GRPO baseline in terms of WUPS score:
>
>     | Model          | Training Data |  NextQA(Open-Ended)
>     |----------------|----------------------|------------------------|
>     | GRPO Baseline  | ACRE-9.2k      | 20.44    |
>     | ACRE (Ours)    | ACRE-9.2k     | 22.58    |
>
>     ACRE achieves a +10.5% relative improvement over GRPO on open-ended generation. This shows that the consistency check during training instills a fundamental reasoning robustness that generalizes beyond the multiple-choice format.

---

> ### Author Response · Authors · 2025-11-22
> **Response to Reviewer 4xWa (Part2/2)**
>
> - **Extending ACRE training to non-MCQ data (Weakness 3).**
>
>   We agree that extending ACRE to non-MCQ data is an important direction. While our main contribution is to resolve the inconsistency issue in dominant MCQ benchmarks, we propose two practical strategies for open-ended data during RL training.
>
>   - **Mixed-reward strategy.**
>     In practice, ACRE does not need to be the only reward mechanism:
>
>     * For **MCQs**: apply the ACRE consistency reward to strictly enforce reasoning alignment.
>     * For **open-ended / math / code**: use outcome-based rewards (e.g., correct answer extraction, unit tests, code execution) or reference-based metrics (ROUGE/BLEU).
>
>     This allows the model to benefit from ACRE’s consistency enforcement without requiring every sample to be in MCQ format.
>
>   - **Automated conversion via distractor generation (preliminary experiment).**
>     To directly apply ACRE’s consistency check to open-ended data, we use an **automated distractor generation** approach. A strong MLLM (GPT-4o-mini) generates plausible but incorrect options (distractors) from the ground-truth answer, temporarily converting an open-ended question into an MCQ for training.
>
>     We ran a preliminary experiment on a 3k subset of LLaVA-Video-178K (open-ended):
>
>     * **Baseline:** standard GRPO using the average of ROUGE-1/2/L as the reward (following Video-R1).
>     * **ACRE:** convert each example to an MCQ with 3 distractors generated by GPT-4o-mini, then apply the ACRE consistency reward.
>
>     The validation results are:
>
>     | Method          | MMVU |
>     |-----------------|-------------------|
>     | GRPO (Baseline) |  0.615   |
>     | ACRE (Ours)     | 0.633    |
>
>     These experiments show that open-ended data can be adapted to the ACRE framework with minimal extra complexity, and without relying on an expensive LLM-as-a-judge loop during RL training. And the preliminary experiments demonstrate its effectiveness over GRPO.

---

> ### Author Response · Authors · 2025-11-26
> **Kindly Remind of Discussion Period**
>
> Dear Reviewer 4xWa,
>
> We would like to thank you again for your detailed reviews. We have explained the applicability of ACRE to open-ended tasks in both training and inference, and the computational overhead in the above response. We have added a discussion subsection in Section 5.
>
> As the discussion period will be closed soon, we would appreciate it if you could let us know if our responses have addressed your concerns and whether you still have any other questions. We would be happy to do any follow-up discussion or address any additional comments.
>
> Again, thank you very much for your attention to our work.

---

### Official Review · Reviewer_SupS · 2025-11-03

**Soundness:** 3
**Presentation:** 3
**Contribution:** 3
**Rating:** 6
**Confidence:** 3

**Summary:**

This paper addresses the "reasoning-answer inconsistency"  that arises when training Multi-modal Large Language Models (MLLMs) with outcome-based reinforcement learning (RL) methods like GRPO. The authors observe that while GRPO improves final answer accuracy, it can decouple the reasoning chain (Chain-of-Thought, CoT) from the answer, leading to situations where the model produces correct reasoning but a wrong answer (CR-WA) or, conversely, flawed reasoning that coincidentally results in a correct answer (WR-CA). To quantify this, the paper introduces two metrics: the CoT Answer Consistency Rate (CACR) and the Option Shuffling Consistency Rate (OSCR). To solve the inconsistency problem, the authors propose Answer-Consistent Reinforcement Learning, a reward-shaping scheme that modifies GRPO. ACRE works by performing an auxiliary consistency check. After the MLLM generates a CoT and an initial answer for a multiple-choice question, the answer options are shuffled. The model is then prompted again with the original reasoning trace and the shuffled options to produce a second answer. A high reward is given only if both the original and post-shuffle answers are correct and identical. This mechanism penalizes reasoning-answer misalignment and discourages reliance on spurious patterns like option ordering.

**Strengths:**

1. The paper's primary strength is its clear diagnosis of the "reasoning-answer inconsistency"  and the cleverness of the proposed solution. The idea of using an auxiliary consistency check based on option shuffling to validate a reasoning trace is highly original. It directly targets the model's reliance on spurious correlations, such as option ordering, which is a known but difficult-to-solve problem.
2. The authors provide convincing empirical evidence for their claims. They first show that standard GRPO training erodes consistency (Tables 1 and 2), validating the premise of their work. They then demonstrate that ACRE not only recovers this consistency but also surpasses the CoT-SFT baseline, all while achieving superior accuracy on the final task.
3. The paper goes beyond just reporting scores. The attention visualization in Figure 3 provides a valuable qualitative insight, suggesting that ACRE learns to attend more to the content of the options and its own reasoning, while GRPO disproportionately attends to the system prompt and option indices.
4. The paper shows that ACRE achieves strong results while being trained on a relatively small dataset (ACRE-9.2k). It even outperforms a model (Video-R1-7B) trained on a much larger dataset (260k samples), highlighting the efficiency of the proposed reward-shaping scheme.

**Weaknesses:**

1. The most significant weakness is that the ACRE framework, as presented, is fundamentally dependent on a multiple-choice question (MCQ) format. The core consistency check relies on the existence of a discrete set of answer options that can be shuffled. This limits the method's direct applicability to many other important tasks, such as open-ended question answering, code generation, or mathematical proof generation, which lack this structure.
2. While the conclusion briefly mentions "alternative rephrasing strategies" like using an LLM to paraphrase the query, this is purely speculative. It's unclear how this would work in practice, how the consistency reward would be calculated (especially without a single ground-truth answer), or what the computational and implementation costs of such a strategy would be.
3. The method inherently requires a second forward pass through the model to generate the post-shuffle answer $\tilde{a}$. The authors report this as a +24% increase in training GPU hours, which, while noted as "acceptable", is a non-trivial overhead.

**Questions:**

1. How do the authors envision adapting ACRE to open-ended generative tasks (e.g., free-form visual question answering or text generation) where there are no explicit "options" to shuffle?
2. Following up on the idea in the conclusion: If an LLM were used to produce "semantically equivalent paraphrases" of a question, how would the consistency-verification reward $r_c$ be formulated? In an open-ended setting, the original answer $a$ and the new answer $\tilde{a}$ would both be free-text. Would this require another LLM-as-Judge to determine if $a$ and $\tilde{a}$ are "consistent" and "correct"? This seems to add significant complexity and potential sources of noise.

---

> ### Author Response · Authors · 2025-11-22
>
> Thanks for your constructive feedback. We handle your concerns as follows.
>
> - **Applicability to open-ended tasks (Weakness 1 & Question 1).**
>
>   We acknowledge the concern regarding the focus on Multiple-Choice Questions (MCQs). However, we believe this focus is justified, and our method also benefits open-ended tasks.
>
>   - **MCQs are ideal for diagnosing inconsistency.**
>     As stated in the paper, a significant portion of current MLLM benchmarks use MCQs because of reliable evaluation and low labeling cost. It is precisely in this format that *reasoning–answer inconsistency* (e.g., positional bias, shortcut learning) is most acute. ACRE is designed to directly target this pathology.
>
>   - **Benefits transfer to open-ended generation.**
>     Although ACRE is trained on MCQs to enforce robust reasoning, the benefits transfer to open-ended settings. By penalizing the reasoning-decision inconsistency, the model learns to ground its answer in evidence rather than heuristics.
>
>     To verify this, we evaluated our ACRE-trained model (RL post-trained only on ACRE-9.2k MCQs) on the open-ended split of NextQA without any further fine-tuning and compared it against the GRPO baseline in terms of WUPS score:
>
>     | Model          | Training Data |  NextQA(Open-Ended)
>     |----------------|----------------------|------------------------|
>     | GRPO Baseline  | ACRE-9.2k      | 20.44    |
>     | ACRE (Ours)    | ACRE-9.2k     | 22.58    |
>
>     **Result:** ACRE achieves a +10.5% relative improvement over GRPO on open-ended generation. This shows that the consistency check during training instills a fundamental reasoning robustness that generalizes beyond the multiple-choice format.
>
> - **Extending ACRE training to non-MCQ data (Weakness 2 & Question 2).**
>
>   We agree that extending ACRE to non-MCQ data is an important direction. While our main contribution is to resolve the inconsistency issue in dominant MCQ benchmarks, we propose two practical strategies for open-ended data during RL training.
>
>   - **Mixed-reward strategy.**
>     In practice, ACRE does not need to be the only reward mechanism:
>
>     * For **MCQs**: apply the ACRE consistency reward to strictly enforce reasoning alignment.
>     * For **open-ended / math / code**: use outcome-based rewards (e.g., correct answer extraction, unit tests, code execution) or reference-based metrics (ROUGE/BLEU).
>
>     This allows the model to benefit from ACRE’s consistency enforcement without requiring every sample to be in MCQ format.
>
>   - **Automated conversion via distractor generation (preliminary experiment).**
>     To directly apply ACRE’s consistency check to open-ended data, we use an **automated distractor generation** approach. A strong MLLM (GPT-4o-mini) generates plausible but incorrect options (distractors) from the ground-truth answer, temporarily converting an open-ended question into an MCQ for training.
>
>     We ran a preliminary experiment on a 3k subset of LLaVA-Video-178K (open-ended):
>
>     * **Baseline:** standard GRPO using the average of ROUGE-1/2/L as the reward (following Video-R1).
>     * **ACRE:** convert each example to an MCQ with 3 distractors generated by GPT-4o-mini, then apply the ACRE consistency reward.
>
>     The validation results are:
>
>     | Method          | MMVU |
>     |-----------------|-------------------|
>     | GRPO (Baseline) |  0.615   |
>     | ACRE (Ours)     | 0.633    |
>
>     These experiments show that open-ended data can be adapted to the ACRE framework with minimal extra complexity, and without relying on an expensive LLM-as-a-judge loop during RL training. And the preliminary experiments demonstrate its effectiveness over GRPO.
>
> - **Computational overhead (Weakness 3).**
>
>    We still believe this is a reasonable cost, especially when viewed from a data-efficiency perspective.
>
>   - **Overhead appears only during training.**
>     The extra cost comes from the second forward pass needed to compute the consistency reward. This affects *training only*. Inference remains a single-pass Chain-of-Thought generation, so there is **no additional latency** for end-users.
>
>   - **Data efficiency outweighs the extra compute.**
>     As highlighted in the paper, ACRE (trained on 9.2k samples) outperforms Video-R1 (trained on 260k samples):
>
>     * Video-R1 processes roughly **28× more data samples**.
>     * ACRE uses about **1.24× compute per sample**.
>
>     Overall, ACRE is therefore *far more compute-efficient* in terms of total GPU-hours required to reach comparable or better performance. The “second pass” effectively serves as a powerful data augmentation mechanism, extracting more robust learning signals from each sample than standard GRPO.
>
>   - **Room for engineering optimization.**
>     As stated in the paper, our current implementation doesn't reuse the KV-cache between the two passes. Since the prompt and reasoning trace are identical, enabling KV-cache reuse could further cut the extra cost to a smaller margin.

---

> ### Author Response · Authors · 2025-11-26
> **Kindly Remind of Discussion Period**
>
> Dear Reviewer SupS,
>
> We would like to thank you again for your detailed reviews. We have explained the applicability of ACRE to open-ended tasks in both training and inference, and the computational overhead in the above response. We have added a discussion subsection in Section 5.
>
> As the discussion period will be closed soon, we would appreciate it if you could let us know if our responses have addressed your concerns and whether you still have any other questions. We would be happy to do any follow-up discussion or address any additional comments.
>
> Again, thank you very much for your attention to our work.

---

### Author Response · Authors · 2025-12-03
**Rebuttal Summary**

Dear Area Chair,

Thank you very much for your time and effort in handling our submission. We thank all reviewers for their insightful and constructive feedback. We would like to provide a brief summary of the reviewers' reactions to our rebuttal and the additional experiments conducted during the discussion phase.

### 1. Update on Ratings

We are pleased to report that **Reviewer Yvee officially raised their score from 4 to 6** (as early as **1.45 a.m., Nov.27, UTC**, as far as we observed.). To the point of 1.45 a.m., Nov.27, the ratings and confidence of the six reviewers are:

| Reviewer | Rating | Confidence |
| :--- | :---: | :---: |
| **Yvee** | **6** | 4 |
| **8rc3** | **6** | 4 |
| **SupS** | **6** | 3 |
| 4xWa | 4 | 3 |
| uJw3 | 4 | 4 |
| tBYd | 2 | 3 |

### 2. Common Strengths Recognized by Reviewers

We are encouraged that the reviewers uniformly recognized the value of our work. Specifically:

* **Insightful Diagnosis:** Reviewers praised the clear diagnosis and quantification of the "reasoning-answer inconsistency" problem in RL fine-tuning (SupS, 4xWa, uJw3, 8rc3).
* **Methodological Elegance:** The proposed ACRE method was highlighted for its elegance and effectiveness, with reviewers noting that the option-shuffling mechanism is a clever and intuitive solution (SupS, 4xWa, Yvee, 8rc3).
* **Convincing Evaluation:** The comprehensive empirical evaluation—particularly the qualitative attention analysis—was acknowledged for providing deep insights into the method's success (SupS, 4xWa, Yvee, 8rc3).

### 3. Response to the Primary Concern: Dependency on MCQs

The most frequently raised concern was the method's reliance on Multiple-Choice Questions (MCQs) (SupS, 4xWa, Yvee, uJw3).

We have clarified that our reward mechanism is designed to replace the reward for GRPO when encountering MCQs when conducting Reinforcement Learning with Verifiable Reward within a mixed-data training setup, the resulting model remains fully capable of inference on non-MCQ tasks. We addressed this by:

1. **Direct Evaluation on Open-Ended Tasks:** We evaluated the model trained via ACRE on the open-ended split of NextQA, observing a **consistent performance gain (+10.5 relative improvement)** over the GRPO baseline.
2. **Distractor Generation:** We added preliminary experiments demonstrating the automatic conversion of open-ended data into MCQ format via distractor generation, further expanding the method's applicability.
3. **Mixed-Reward Strategy:** We clarified that ACRE integrates seamlessly with other reward functions, allowing for training on mixed data formats.

*Note: These responses fully addressed the concerns of Yvee (High Confidence), leading to the **score being increased to 6**. We believe these explanations also resolve similar concerns from other reviewers, including 4xWa (High Confidence), who has expressed a clear willingness to **raise the score** if the concerns are addressed.*

### 4. Response to Other Concerns

* **Model Scaling (Yvee, uJw3):** While our experiments focus on 7B-scale models, this aligns with standards in recent top-tier publications (e.g., NeurIPS, ICCV). Training RL on 30B+ models requires computational resources beyond typical academic settings.
* **Why Multimodal? (Yvee, tBYd)** We clarified that the "reasoning-answer inconsistency" is significantly more severe in Multimodal LLMs compared to text-only LLMs, justifying our specific focus on the multimodal domain.
* **Computational Costs (SupS, 4xWa):** While ACRE involves additional computational steps, it is highly data-efficient. We achieve results comparable to baselines that utilize roughly **28× more data samples**, making the trade-off highly favorable.

We believe that our responses during the discussion phase, combined with the updates in the revised manuscript (all revisions are highlighted in $\color{blue}{\text{blue}}$), have comprehensively addressed all the concerns raised. We reiterate our gratitude for the constructive feedback and remain open to any further suggestions that could help improve this work.

Best regards,

Authors

---

### Meta-Review · Area_Chair_vKh8 · 2026-01-06

**Summary:**

The majority of the concerns for this paper focused on two aspects:
1. The limited scope of the ACRE framework, as this works for the Multiple-choice question format. It is non trivial to extend it to the other tasks (SupS, 4xWa, Yvee, uJw3, 8rc3)
2. The second forward pass introduced significant training overhead but only brings limited performance gain (SupS, 4xWa)

There are other concerns related to the presentation (tBYd), the motivation and the experiment (8rc3)

**Reviewer Concerns:**

To judge whether this paper is deemed for acceptance, I think the first two concerns need to be fully addressed.

**The limited scope of the ACRE framework:**

The author rebuttal mentioned the following evidence to support the proposed framework
1. MCQs are ideal for diagnosing inconsistency.
2. Benefits transfer to open-ended generation.
3. Extending ACRE training to non-MCQ data with Mixed-reward strategy and Automated conversion via distractor generation.

I agree with the author that the MCQs are ideal for diagnosing the inconsistency, which is a good standing point for the proposed approach. However, comparing with the free-form QA format, the MCQs are restricted. The author demonstrated the NextQA performance to solidify the claim for benefiting the transfer to open-ended generation. However, only NextQA improvement might not be comprehensive enough to justify the transferability to the open-ended generation. More explicit conceptual reasoning (other than empirical results) might need to be presented to justify why would the ACRE framework benefits the open-ended generation. The evidence for supporting this claim might not be enough. For those two apporaches for extending the MCQ to non-MCQ data: 1. mixed-reward strategy: I agree with the author that the ACRE could be used as an additional reward signals for model training. But for the automated conversion strategy, I am not fully persuaded that using ACRE to train on the automated converted MCQs is more beneficial than using GRPO to train on the original open-ended task format.

Given the above, I have not been fully persuaded that the limited scope of the ACRE framework is fully addressed.

**Significant training overhead with limited performance improvement**

The author rebuttal comments that
1. Overhead appears only during training.
2. Data efficiency outweighs the extra compute.

For the overhead appears only during the training, I agree with this claim. However, I think this answer is orthogonal to the reviewer's concern. The reviewer's concern is mainly around whether the additional computation cost during the training is truly necessary for the limited performance improvement. I don't think the author answered this during the rebuttal. For the data efficiency statement, the author comments that Video-R1 processes roughly 28× more data samples and ACRE uses about 1.24× compute per sample, so that the ACRE is compute-efficiency (CE-win) than GRPO. Frankly speaking, this claim is overclaimed and slightly weak. I think the author might be able to claim CE win on the MCQ tasks, but need a more thorough study on that (e.g. scaling to larger scale) Directly claiming 'far more compute-efficient' might not be correct.

**Reviewer Scores:**

As I mentioned above, I think the main concerns about this paper (the limited scope and the extra comptuation costs) are not fully addressed. I think this is an interesting paper, but the ACRE framework has its own limitation which might not pass the acceptance bar.

---

### Decision · Program_Chairs · 2026-01-26

Reject